# Accelerating SGD for Highly Ill-Conditioned Huge-Scale Online Matrix Completion

**Gavin Zhang**
University of Illinois at Urbana–Champaign
`jialun2@illinois.edu`

**Hong-Ming Chiu**
University of Illinois at Urbana–Champaign
`hmchiu2@illinois.edu`

**Richard Y. Zhang**
University of Illinois at Urbana–Champaign
`ryz@illinois.edu`

## Abstract

The matrix completion problem seeks to recover a $d \times d$ ground truth matrix of low rank $r \ll d$ from observations of its individual elements. Real-world matrix completion is often a huge-scale optimization problem, with $d$ so large that even the simplest full-dimension vector operations with $O(d)$ time complexity become prohibitively expensive. Stochastic gradient descent (SGD) is one of the few algorithms capable of solving matrix completion on a huge scale, and can also naturally handle streaming data over an evolving ground truth. Unfortunately, SGD experiences a dramatic slow-down when the underlying ground truth is ill-conditioned; it requires at least $O(\kappa \log(1/\epsilon))$ iterations to get $\epsilon$-close to ground truth matrix with condition number $\kappa$. In this paper, we propose a preconditioned version of SGD that preserves all the favorable practical qualities of SGD for huge-scale online optimization while also making it agnostic to $\kappa$. For a symmetric ground truth and the Root Mean Square Error (RMSE) loss, we prove that the preconditioned SGD converges to $\epsilon$-accuracy in $O(\log(1/\epsilon))$ iterations, with a rapid linear convergence rate as if the ground truth were perfectly conditioned with $\kappa = 1$. In our experiments, we observe a similar acceleration for item-item collaborative filtering on the MovieLens25M dataset via a pair-wise ranking loss, with 100 million training pairs and 10 million testing pairs. [See supporting code at https://github.com/Hong-Ming/ScaledSGD.]

## 1 Introduction

The matrix completion problem seeks to recover an underlying $d \times d$ ground truth matrix $M$ of low rank $r \ll d$ from observations of its individual matrix elements $M_{i,j}$. The problem appears most prominently in the context of collaborative filtering and recommendation system, but also numerous other applications. In this paper, we focus on the *symmetric* and *positive semidefinite* variant of the problem, in which the underlying matrix $M$ can be factored as $M = ZZ^T$ where the factor matrix $Z$ is $d \times r$, though that our methods have natural extensions to the nonsymmetric case. We note that the symmetric positive semidefinite variant is actually far more common in collaborative filtering, due to the prevalence of *item-item* models, which enjoy better data (most platforms contain several orders of magnitude more users than items) and more stable recommendations (the similarity between items tends to change slowly over time) than user-user and user-item models.

For the full-scale, online instances of matrix completion that arise in real-world collaborative filtering, stochastic gradient descent or SGD is the only viable algorithm for learning the underlying matrix $M$. The basic idea is to formulate a candidate matrix of the form $XX^T$ with respect to a

36th Conference on Neural Information Processing Systems (NeurIPS 2022).

learned factor matrix $X \in \mathbb{R}^{d \times r}$, and to minimize a cost function of the form $\phi(XX^T - M)$. Earlier work used the root mean square error (RMSE) loss $\|XX^T - M\|_F^2 = \sum_{i,j} (XX^T - M)_{i,j}^2$, though later work have focused on pairwise losses like the BPR [1] that optimize for ordering and therefore give better recommendations. For the RMSE loss, the corresponding SGD iterations with (rescaled) learning rate $\alpha > 0$ reads

$$x_{i,+} = x_i - \alpha \cdot \left(x_i^T x_j - M_{ij}\right) x_j, \quad x_{j,+} = x_j - \alpha \cdot \left(x_i^T x_j - M_{ij}\right) x_i, \tag{1}$$

where $M_{ij}$ is the sampled $(i,j)$-th element of the ground truth matrix $M$, and $x_i, x_j$ and $x_{i,+}, x_{j,+}$ denote the $i$-th and $j$-th rows of the current iterate $X_t$ and new iterate $X_{t+1}$. Pairwise losses like the BPR can be shown to have a similar update equation over three rows of $X$ [1]. Given that only two or three rows of $X$ are accessed and updated at any time, SGD is readily accessible to massive parallelization and distributed computing. For very large values of $d$, the update equation (1) can be run by multiple workers in parallel without locks, with vanishing probability of collision [2]. The blocks of $X$ that are more frequently accessed together can be stored on the same node in a distributed memory system.

Unfortunately, the convergence rate of SGD can sometimes be extremely slow. One possible explanation, as many recent authors have pointed out [3–6], is that matrix factorization models are very sensitive to ill-conditioning of the ground truth matrix $M$. The number of SGD iterations grows at least linearly the condition number $\kappa$, which here is defined as the ratio between the largest and the $r$-th largest singular values of $M$. Ill-conditioning causes particular concern because most real-world data are ill-conditioned. In one widely cited study [7], it was found that the dominant singular value accounts for only $\approx 80\%$ prediction accuracy, with diversity of individual preferences making up the remainder ill-conditioned singular values. Cloninger et al. [8] notes that there are certain applications of matrix completion that have condition numbers as high as $\kappa = 10^{15}$.

This paper is inspired by a recent full-batch gradient method called ScaledGD [4, 9] and a closely related algorithm PrecGD [5] in which gradient descent is made immune to ill-conditioning in the ground truth by right-rescaling the full-batch gradient by the matrix $(X^T X)^{-1}$. Applying this same strategy to the SGD update equation (1) yields the row-wise updates

$$x_{i,+} = x_i - \alpha \cdot \left(x_i^T x_j - M_{ij}\right) P x_j, \quad x_{j,+} = x_j - \alpha \cdot \left(x_i^T x_j - M_{ij}\right) P x_i, \tag{2a}$$

in which we *precompute and cache* the preconditioner $P = (X^T X)^{-1}$ ahead of time[1], and update it after the iteration as

$$P_+ = (P^{-1} + x_{i,+} x_{i,+}^T + x_{j,+} x_{j,+}^T - x_i x_i^T - x_j x_j^T)^{-1} \tag{2b}$$

by making four calls to the Sherman–Morrison rank-1 update formula

$$(P^{-1} + uu^T)^{-1} = P - \frac{Puu^T P}{1 + u^T P u}, \qquad (P^{-1} - uu^T)^{-1} = P + \frac{Puu^T P}{1 - u^T P u}.$$

This way, the rescaled update equations use just $O(r^2)$ arithmetic operations, which for modest values of $r$ is only marginally more than the $O(r)$ cost of the unscaled update equations (1). Indeed, the nearest-neighbor algorithms inside most collaborative filters have exponential complexity with respect to the latent dimensionality $r$, and so are often implemented with $r$ small enough for (1) and (2) to have essentially the same runtime. Here, we observe that the rescaled update equations (2) preserve essentially all of the practical advantages of SGD for huge-scale, online optimization: it can also be run by multiple workers in parallel without locks, and it can also be easily implemented over distributed memory. The only minor difference is that separate copies of $P$ should be maintained by each worker, and resynchronized once differences grow large.

**Contributions**  In this paper, we provide a rigorous proof that the rescaled update equations (1), which we name ScaledSGD, become *immune* to the effects of ill-conditioning in the underlying ground truth matrix. For symmetric matrix completion under the root mean squared error (RMSE) loss function, regular SGD is known to have an iteration count of $O(\kappa^4 \cdot dr \log(d/\epsilon))$ within a local neighborhood of the ground truth [10]. This figure is optimal in the dimension $d$, the rank

---

[1]For an initialization, if the $d$ rows of $X_0$ are selected from the unit Gaussian as in $x_1, \ldots, x_d \sim \mathcal{N}(0, \sigma^2 I_r)$, then we can simply set $P_0 = \sigma^2 I$ without incurring the $O(d)$ time needed in explicitly computing $P_0 = (X_0^T X_0)^{-1}$.

$r$, and the final accuracy $\epsilon$, but suboptimal by four exponents with respect to condition number $\kappa$. In contrast, we prove for the same setting that ScaledSGD attains an optimal convergence rate, converging to $\epsilon$-accuracy in $O(dr \log(d/\epsilon))$ iterations for all values of the condition number $\kappa$. In fact, our theoretical result predicts that ScaledSGD converges as if the ground truth matrix is perfectly conditioned, with a condition number of $\kappa = 1$.

At first sight, it appears quite natural that applying the ScaledGD preconditioner to SGD should result in accelerated convergence. However, the core challenge of stochastic algorithms like SGD is that each iteration can have substantial *variance* that "drown out" the expected progress made in the iteration. In the case of ScaledSGD, a rough analysis would suggest that the highly ill-conditioned preconditioner should improve convergence in expectation, but at the cost of dramatically worsening the variance.

Surprisingly, we find in this paper that the specific scaling $(X^T X)^{-1}$ used in ScaledSGD not only does not worsen the variance, but in fact *improves* it. Our key insight and main theoretical contribution is Lemma 4, which shows that the same mechanism that allows ScaledGD to converge faster (compared to regular GD) also allows ScaledSGD to enjoy reduced variance (compared to regular SGD). In fact, it is this effect of variance reduction that is responsible for most ($\kappa^3$ out of $\kappa^4$) of our improvement over the previous state-of-the-art. It turns out that a careful choice of preconditioner can be used as a mechanism for *variance reduction*, while at the same time also fulfilling its usual, classical purpose, which is to accelerate convergence in expectation.

**Related work**  Earlier work on matrix completion analyzed a convex relaxation of the original problem, showing that nuclear norm minimization can recover the ground truth from a few incoherent measurements [11–15]. This approach enjoys a near optimal sample complexity but incurs an $O(d^3)$ per-iteration computational cost, which is prohibitive for a even moderately large $d$. More recent work has focused more on a nonconvex formulation based on Burer and Monteiro [16], which factors the optimization variable as $M = XX^T$ where $X \in \mathbb{R}^{d \times r}$ and applies a local search method such as alternating-minimization [17–20], projected gradient descent [21, 22] and regular gradient descent [23–26]. A separate line of work [27–35] focused on global properties of nonconvex matrix recovery problems, showing that the problem has no spurious local minima if sampling operator satisfies certain regularity conditions such as incoherence or restricted isometry.

The convergence rate of SGD has been well-studied for general classes of functions [36–39]. For matrix completion in particular, Jin et al. [10] proved that SGD converges towards an $\epsilon$-accurate solution in $O(d\kappa^4 \log(1/\epsilon))$ iterations where $\kappa$ is the condition number of $M$. Unfortunately, this quartic dependence on $\kappa$ makes SGD extremely slow and impractical for huge-scale applications.

This dramatic slow down of gradient descent and its variants caused by ill-conditioning has become well-known in recent years. Several recent papers have proposed full-batch algorithms to overcome this issue [9, 40, 41], but these methods cannot be used in the huge-scale optimization setting where $d$ is so large that even full-vector operations with $O(d)$ time complexity are too expensive. As a deterministic full-batch method, ScaledGD [9] requires a projection onto the set of incoherent matrices at every iteration in order to maintain rapid convergence. Instead our key finding here is that the stochasticity of SGD alone is enough to keep the iterates as incoherent as the ground truth, which allows for rapid progress to be made. The second-order method proposed in [41] costs at least $O(d)$ per-iteration and has no straightforward stochastic analog. PrecGD [5] only applies to matrices that satisfies matrices satisfying the restricted isometry property, which does not hold for matrix completion.

## 2  Background: Linear convergence of SGD

In our theoretical analysis, we restrict our attention to symmetric matrix completion under the root mean squared error (RMSE) loss function. Our goal is to solve the following nonconvex optimization

$$\min_{X \in \mathbb{R}^{d \times r}} f(X) \stackrel{\text{def}}{=} \|XX^T - ZZ^T\|_F^2 \qquad \text{where } Z = [z_1, z_2, \dots, z_n]^T \in \mathbb{R}^{d \times r} \qquad (3)$$

in which we assume that the $d \times d$ ground truth $ZZ^T \succeq 0$ matrix is exactly rank-$r$, with a finite *condition number*

$$\kappa \stackrel{\text{def}}{=} \lambda_{\max}(ZZ^T)/\lambda_r(ZZ^T) = \lambda_{\max}(Z^T Z)/\lambda_{\min}(Z^T Z) < \infty. \qquad (4)$$

In order to be able to reconstruct $ZZ^T$ from a small number of measurements, we will also need to assume that the ground truth has small *coherence* [42]

$$\mu \overset{\text{def}}{=} \frac{d}{r} \cdot \max_{1 \le i \le d} \|e_i^T Z (Z^T Z)^{-1/2}\|^2. \tag{5}$$

Recall that $\mu$ takes on a value from 1 to $d/r$, with the smallest achieved by dense, orthonormal choices of $Z$ whose rows all have magnitudes of $1/\sqrt{d}$, and the largest achieved by a ground truth $ZZ^T$ containing a single nonzero element. Assuming incoherence $\mu = O(1)$ with respect to $d$, it is a well-known result that all $d^2$ matrix elements of $ZZ^T$ can be perfectly reconstructed from just $O(dr \log d)$ random samples of its matrix elements [12, 43].

This paper considers solving (3) in the huge-scale, online optimization setting, in which individual matrix elements of the ground truth $(ZZ^T)_{i,j} = z_i^T z_j$ are revealed one-at-a-time, uniformly at random with replacement, and that a current iterate $X$ is continuously updated to streaming data. We note that this is a reasonably accurate model for how recommendation engines are tuned to user preferences in practice, although the uniformity of random sampling is admittedly an assumption made to ease theoretical analysis. Define the stochastic gradient operator as

$$SG(X) = 2d^2 \cdot (x_i^T x_j - z_i^T z_j)(e_i x_j^T + e_j x_i^T) \quad \text{where } (i,j) \sim \text{Unif}([d] \times [d]),$$

where $x_i, x_j \in \mathbb{R}^r$ are the $i$-th and $j$-th rows of $X$, and the scaling $d^2$ is chosen that, over the randomness of the sampled index $(i, j)$, we have exactly $\mathbb{E}[SG(X)] = \nabla f(X)$. Then, the classical online SGD algorithm can be written as

$$X_{t+1} = X_t - \alpha SG(X_t) \qquad \text{where } \alpha > 0. \tag{SGD}$$

Here, we observe that a single iteration of SGD coincides with full-batch gradient descent in expectation, as in $\mathbb{E}[X_{t+1}|X_t] = X_t - \alpha \nabla f(X_t)$. Therefore, assuming that bounded deviations and bounded variances, it follows from standard arguments that the behavior of many iterations of SGD should concentrate about that of full-batch gradient descent $X_{t+1} = X_t - \alpha \nabla f(X_t)$.

Within a region sufficiently close to the ground truth, full-batch gradient descent is well-known to converge at a linear rate to the ground truth [23, 44]. Within this same region, Jin et al. [10] proved that SGD also converges linearly. For an incoherent ground truth with $\mu = O(1)$, they proved that SGD with an aggressive choice of step-size is able to recover the ground truth to $\epsilon$-accuracy $O(\kappa^4 dr \log(d/\epsilon))$ iterations, with each iteration costing $O(r)$ arithmetic operations and selecting 1 random sample. This iteration count is optimal with respect to $d$, $r$, and $1/\epsilon$, although its dependence on $\kappa$ is a cubic factor (i.e. a factor of $\kappa^3$ worse than full-batch gradient descent's figure of $O(\kappa \log(1/\epsilon))$, which is itself already quite bad, given that $\kappa$ in practice can readily take on values of $10^3$ to $10^6$.

**Theorem 1** (Jin, Kakade, and Netrapalli [10]). *For $Z \in \mathbb{R}^{d \times r}$ with $\sigma_{\max}(Z) = 1$ and $f(X) = \|XX^T - ZZ^T\|_F^2$ and $h_i(X) = \|e_i^T X\|^2$, define the following*

$$f_{\max} \overset{\text{def}}{=} \left(\frac{1}{10\kappa}\right)^2, \qquad h_{\max} \overset{\text{def}}{=} 20 \cdot \kappa^2 \cdot \frac{\mu r}{d}.$$

*For an initial point $X_0 \in \mathbb{R}^{d \times r}$ that satifies $f(X_0) \le \frac{1}{2} f_{\max}$ and $\max_i h_i(X_0) \le \frac{1}{2} h_{\max}$, there exists some constant $c$ such that for any learning rate $\alpha < c \cdot (\kappa \cdot h_{\max} \cdot d^2 \log d)^{-1}$, with probability at least $1 - T/d^{10}$, we will have for all $t \le T$ iterations of SGD that*

$$f(X_t) \le \left(1 - \frac{\alpha}{2 \cdot \kappa}\right)^t \cdot f_{\max}, \qquad \max_i h_i(X_t) \le h_{\max}.$$

The reason for Theorem 1's additional $\kappa^3$ dependence beyond full-batch gradient descent is due to its need to maintain *incoherence* in its iterates. Using standard techniques on martingale concentration, one can readily show that SGD replicates a single iteration of full-batch gradient descent over an epoch of $d^2$ iterations. This results in an iteration count $O(\kappa \cdot d^2 \log(1/\epsilon))$ with an optimal dependence on $\kappa$, but the entire matrix is already fully observed after collecting $d^2$ samples. Instead, Jin et al. [10] noted that the *variance* of SGD iterations is controlled by the step-size $\alpha$ times the maximum coherence $\mu_X = \frac{d}{r} \cdot \max_{i,t} \|e_i^T X_t\|^2$ over the iterates $X_t, X_{t-1}, \ldots, X_0$. If the iterates can be kept incoherent with $\mu_X = O(1)$, then SGD with a more aggressive step-size will reproduce an iteration of full-batch gradient descent after an epoch of just $O(dr \log d)$ iterations.

The main finding in Jin et al. [10]'s proof of Theorem 1 is that the stochasticity of SGD is enough to keep the iterates incoherent. This contrasts with full-batch methods at the time, which required an added regularizer [20, 30, 45] or an explicit projection step [9]. (As pointed out by a reviewer, it was later shown by Ma et al. [46] that full-batch gradient descent is also able to maintain incoherence without a regularizer nor a projection.) Unfortunately, maintaining incoherence requires shrinking the step-size by a factor of $\kappa$, and the actual value of $\mu_X$ that results is also a factor of $\kappa^2$ worse than the original coherence $\mu$ of the ground truth $Z$. The resulting iteration count $O(\kappa^4 \cdot dr \log(d/\epsilon))$ is made optimal with respect to $d$, $r$, and $1/\epsilon$, but only at the cost of worsening its the dependence on the condition number $\kappa$ by another *three exponents*.

Finally, the quality of the initial point $X_0$ also has a dependence on the condition number $\kappa$. In order to guarantee linear convergence, Theorem 1 requires $X_0$ to lie in the neighborhood $\|X_0 X_0^T - ZZ^T\|_F < \lambda_{\min}(Z^T Z) = O(\kappa^{-1})$. This dependence on $\kappa$ is optimal, because full-batch gradient descent must lose its ability to converge linearly in the limit $\kappa \to \infty$ [5, 6]. However, the leading constant can be very pessimistic, because the theorem must formally exclude spurious critical points $X_{\mathrm{spur}}$ that have $\nabla f(X_{\mathrm{spur}}) = 0$ but $f(X_{\mathrm{spur}}) > 0$ in order to be provably correct. In practice, it is commonly observed that SGD converges globally, starting from an arbitrary, possibly random initialization [30], at a linear rate that is consistent with local convergence theorems like Theorem 1. It is now commonly argued that gradient methods can escape saddle points with high probability [47], and so their performance is primarily dictated by local convergence behavior [48, 49].

## 3    Proposed algorithm and main result

Inspired by a recent full-batch gradient method called ScaledGD [4, 9] and a closely related algorithm PrecGD [5], we proposed the following algorithm

$$X_{t+1} = X_t - \alpha SG(X_t)(X_t^T X_t)^{-1} \qquad \text{where } \alpha > 0. \qquad \text{(ScaledSGD)}$$

As we mentioned in the introduction, the preconditioner $P = (X^T X)^{-1}$ can be precomputed and cached in a practical implementation, and afterwards efficiently updated using the Sherman–Morrison formula. The per-iteration cost of ScaledSGD is $O(r^2)$ arithmetic operations and 1 random sample, which for modest values of $r$ is only marginally more than the cost of SGD.

Our main result in this paper is that, with a region sufficiently close to the ground truth, this simple rescaling allows ScaledSGD to converge linearly to $\epsilon$-accuracy $O(dr \log(d/\epsilon))$ iterations, with no further dependence on the condition number $\kappa$. This iteration count is optimal with respect to $d$, $r$, $1/\epsilon$, and $\kappa$, and in fact matches SGD with a *perfectly conditioned* ground truth $\kappa = 1$. In our numerical experiments, we observe that ScaledSGD converges globally from a random initialization at the same rate as SGD as if $\kappa = 1$.

**Theorem 2** (Main). *For $Z \in \mathbb{R}^{d \times r}$ with $\sigma_{\max}(Z) = 1$ and $f(X) = \|XX^T - ZZ^T\|_F^2$ and $g_i(X) = e_i^T X(X^T X)^{-1} X^T e_i$, select a radius $\rho < 1/2$ and set*

$$f_{\max} \stackrel{\text{def}}{=} \left(\frac{\rho}{\kappa}\right)^2, \qquad g_{\max} \stackrel{\text{def}}{=} \frac{2^4}{(1-2\rho)^2} \cdot \frac{\mu r}{d}.$$

*For an initial point $X_0 \in \mathbb{R}^{d \times r}$ that satifies $f(X_0) \le \frac{1}{2} f_{\max}$ and $\max_i g_i(X_0) \le \frac{1}{2} g_{\max}$, there exists some constant $c$ such that for any learning rate $\alpha < c \cdot [(g_{\max} + \rho) \cdot d^2 \log d]^{-1}$, with probability at least $1 - T/d^{10}$, we will have for all $t \le T$ iterations of ScaledSGD that:*

$$f(X_t) \le \left(1 - \frac{\alpha}{2}\right)^t \cdot f_{\max}, \qquad \max_i g_i(X_t) \le g_{\max}.$$

Theorem 2 eliminates all dependencies on the condition number $\kappa$ in Theorem 1 except for the quality of the initial point, which we had already noted earlier as being optimal. Our main finding is that it is possible to maintain incoherence while making aggressive step-sizes towards a highly ill-conditioned ground truth $ZZ^T$. In fact, Theorem 2 says that, with high probability, the maximum coherence $\mu_X$ over of any iterate $X_t$ will only be a *mild constant factor* of $\approx 16$ times worse than the coherence $\mu$ of the ground truth $ZZ^T$. This is particularly surprising in view of the fact that every iteration of ScaledSGD involves inverting a potentially highly ill-conditioned matrix $(X^T X)^{-1}$. In

contrast, even without inverting matrices, Theorem 1 says that SGD is only able to keep $\mu_X$ within a factor of $\kappa^2$ of $\mu$, and only by shrinking the step-size $\alpha$ by another factor of $\kappa$.

However, the price we pay for maintaining incoherence is that the quality of the initial point $X_0$ now gains a dependence on dimension $d$, in addition to the condition number $\kappa$. In order to guarantee fast linear convergence independent of $\kappa$, Theorem 2 requires $X_0$ to lie in the neighborhood $\|X_0 X_0^T - ZZ^T\|_F < \mu r \lambda_{\min}(Z^T Z)/d = (\kappa d)^{-1}$, so that $\rho$ can be set to be the same order of magnitude as $g_{\max}$. In essence, the "effective" condition number of the ground truth has been worsened by another factor of $d$. This shrinks the size of our local neighborhood by a factor of $d$, but has no impact on the convergence rate of the resulting iterations.

In the limit that $\kappa \to \infty$ and the search rank $r$ becomes *overparameterized* with respect to the true rank $r^\star < r$ of $ZZ^T$, both full-batch gradient descent and SGD slows down to a *sublinear* convergence rate, in theory and in practice [5, 6]. While Theorem 2 is no longer applicable, we observe in our numerical experiments that ScaledSGD nevertheless maintains its fast linear convergence rate as if $\kappa = 1$. Following PrecGD [5], we believe that introducing a small identity perturbation to the scaling matrix of ScaledSGD, as in $(X^T X + \eta I)^{-1}$ for some $\eta \approx \sqrt{f(X)}$, should be enough to rigorously extend Theorem 2 to the overparameterized regime. We leave this extension as future work.

# 4 Key ideas for the proof

We begin by explaining the mechanism by which SGD slows down when converging towards an ill-conditioned ground truth. Recall that

$$\mathbb{E}[SG(X)] = \mathbb{E}[2d^2 \cdot (XX^T - ZZ^T)_{i,j} \cdot (e_i e_j^T + e_j e_i^T)X] = 4(XX^T - ZZ^T)X = \nabla f(X).$$

As $XX^T$ converges towards an ill-conditioned ground truth $ZZ^T$, the factor matrix $X$ must become progressively ill-conditioned, with

$$\lambda_{\min}(X^T X) = \lambda_r(XX^T) \leq \lambda_r(ZZ^T) + \|XX^T - ZZ^T\|_F \leq \frac{1+\rho}{\kappa}.$$

Therefore, it is possible for components of the error vector $XX^T - ZZ^T$ to become "invisible" by aligning within the ill-conditioned subspaces of $X$. As SGD progresses towards the solution, these ill-conditioned subspaces of $X$ become the slowest components of the error vector to converge to zero. On the other hand, the maximum step-size that can be taken is controlled by the most well-conditioned subspaces of $X$. A simple idea, therefore, is to rescale the ill-conditioned components of the gradient $\nabla f(X)$ in order to make the ill-conditioned subspaces of $X$ more "visible".

More concretely, define the local norm of the gradient as $\|\nabla f(X)\|_X = \|\nabla f(X)(X^T X)^{1/2}\|_F$ and its corresponding dual norm as $\|\nabla f(X)\|_X^* = \|\nabla f(X)(X^T X)^{-1/2}\|_F$. It has long been known (see e.g. [23, 44]) that rescaling the gradient yields

$$\|\nabla f(X)\|_X^* \overset{\text{def}}{=} \|4(XX^T - ZZ^T)X(X^T X)^{-1/2}\|_F = 4\cos\theta \cdot \|XX^T - ZZ^T\|_F,$$

where $\theta$ is the angle between the error vector $XX^T - ZZ^T$ and the linear subspace $\{XY^T + YX^T : Y \in \mathbb{R}^{d\times r}\}$. This insight immediately suggests an iteration like $X_+ = X - \alpha\nabla f(X)(X^T X)^{-1}$. In fact, the gradients of $f$ have some Lipschitz constant $L$, so

$$f(X_+) \leq f(X) - \alpha\langle \nabla f(X), \nabla f(X)(X^T X)^{-1}\rangle + \frac{L}{2}\alpha^2\|\nabla f(X)(X^T X)^{-1}\|_F^2,$$

$$\leq f(X) - \alpha(\|\nabla f(X)\|_X^*)^2 + \frac{L_X}{2}\alpha^2(\|\nabla f(X)\|_X^*)^2,$$

$$\leq \left[1 - \alpha \cdot 8\cos^2\theta\right] f(X) \qquad \text{for } \alpha \leq 1/L_X.$$

However, a naive analysis finds that $L_X = L/\lambda_{\min}(X^T X) \approx L \cdot \kappa$, and this causes the step-size to shrink by a factor of $\kappa$. The main motivating insight behind ScaledGD [4, 9] and later PrecGD [5] is that, with a finer analysis, it is possible to prove Lipschitz continuity under a local change of norm.

**Lemma 3** (Function descent). *Let $X, Z \in \mathbb{R}^{n\times r}$ satisfy $\|XX^T - ZZ^T\|_F \leq \rho\lambda_{\min}(Z^T Z)$ where $\rho < 1/2$. Then, the function $f(X) = \|XX^T - ZZ^T\|_F^2$ satisfies*

$$f(X + V) \leq f(X) + \langle \nabla f(X), V\rangle + \frac{L_X}{2}\|V\|_X^2, \qquad (\|\nabla f(X)\|_X^*)^2 \geq 13 \cdot f(X)$$

*for all $\|V\|_X \leq C \cdot \sqrt{f(X)}$ with $L_X = 6 + 8C + 2C^2 = O(1 + C^2)$.*

This same idea can be "stochastified" in a straightforward manner. Conditioning on the current iterate $X$, then the new iterate $X_+ = X - \alpha SG(X)(X^T X)^{-1}$ has expectation

$$\mathbb{E}[f(X_+)] \leq f(X) - \alpha \langle \nabla f(X), \mathbb{E}[SG(X)(X^T X)^{-1}] \rangle + \alpha \frac{L_X}{2} \mathbb{E}[(\|SG(X)\|_X^*)^2].$$

The linear term evaluates as $\mathbb{E}[SG(X)(X^T X)^{-1}] = \nabla f(X)(X^T X)^{-1}$, while the quadratic term is

$$\mathbb{E}[(\|SG(X)\|_X^*)^2] \leq \sum_{i,j} 4d^2 \cdot (XX^T - ZZ^T)_{i,j}^2 \cdot 4 \max_i(\|e_i^T X\|_X^*)^2 = 16 \cdot f(X) \cdot \max_i g_i(X),$$

where $g_i(X) = e_i^T X (X^T X)^{-1} X^T e_i = (\|e_i^T X\|_X^*)^2$. Combined, we obtain geometric convergence

$$\mathbb{E}[f(X_+)] \leq \left(1 - \alpha \cdot 8 \cos^2 \theta \right) f(X) \qquad \text{for } \alpha = O(g_{\max}^{-1} \cdot d^{-2}). \tag{6}$$

We see that the step-size depends crucially on the incoherence $g_i(X) \leq g_{\max}$ of the current iterate. If the current iterate $X$ is incoherent with $g_{\max} = O(1/d)$, then a step-size of $\alpha = O(1/d)$ is possible, resulting in convergence in $O(dr \log(d/\epsilon))$ iterations, which can be shown using standard martingale techniques [10]. But if the current iterate is $g_{\max} = O(1)$, then only a step-size of $\alpha = O(1/d^2)$ is possible, which forces us to compute $d^2$ iterations, thereby obviating the need to complete the matrix in the first place.

Therefore, in order for prove rapid linear convergence, we need to additionally show that with high probability, the coherence $g_k(X) = (\|e_k^T X\|_X^*)^2$ remains $O(1)$ throughout ScaledGD iterations. This is the most challenging part of our proof. Previous methods that applied a similar scaling to full-batch GD [9] required an explicit projection onto the set of incoherent matrices at each iteration. Applying a similar projection to ScaledSGD will take $O(d)$ time, which destroys the scalability of our method. On the other hand, Jin et al. [10] showed that the randomness in SGD is enough to keep the coherence of the iterates within a factor of $\kappa^2$ times worse than the coherence of the ground truth, and only by a step-size of at most $\alpha = O(\kappa^{-1})$.

Surprisingly, here we show that the randomness in ScaledSGD is enough to keep the coherence of the iterates with a *constant factor* of the coherence the ground truth, using a step-size with no dependence on $\kappa$. The following key lemma is the crucial insight of our proof. First, it says that function $g_k(X)$ satisfies a "descent lemma" with respect to the local norm $\| \cdot \|_X^*$. Second, and much more importantly, it says that descending $g_k(X)$ along the scaled gradient direction $\nabla f(X)(X^T X)^{-1}$ incurs a linear decrement $\frac{1-2\rho}{1-\rho} g_k(X)$ with no dependence of the condition number $\kappa$. This is in direct analogy to the function value decrement in (6), which has no dependence on $\kappa$, and in direct contrast to the proof of Jin et al. [10], which is only able to achieve a decrement of $(8/\kappa) g_k(X)$ due to the lack of rescaling by $(X^T X)^{-1}$.

**Lemma 4** (Coherence descent). *Let $g_k(X) = e_k^T X (X^T X)^{-1} X^T e_k$. Under the same conditions as Lemma 3, we have*

$$g_k(X + V) \leq g_k(X) + \langle V, \nabla g_k(X) \rangle + \frac{5(\|V\|_X^*)^2}{1 - 2\|V\|_X^*},$$

$$\langle \nabla g_k(X), \nabla f(X)(X^T X)^{-1} \rangle \geq \left[ \frac{1 - 2\rho}{1 - \rho} g_k(X) - \frac{1}{1 - \rho} \sqrt{g_k(X) g_k(Z)} \right].$$

Conditioning on $X$, we have for the search direction $V = SG(X)(X^T X)^{-1}$ and $X_+ = X + V$

$$\mathbb{E}[g_k(X_+)] \leq g_k(X) - \alpha \langle \nabla g_k(X), \mathbb{E}[V] \rangle + \alpha^2 \cdot \mathbb{E}\left[ \frac{(\|V\|_X^*)^2}{1 - 2\|V\|_X^*} \right]$$

$$\leq \left(1 - \frac{1 - 2\rho}{1 - \rho} \alpha \right) g_k(X) + \alpha \cdot \frac{1}{1 - \rho} \cdot \sqrt{g_k(X) g_k(Z)} + \alpha^2 \cdot \frac{\mathbb{E}\left[(\|V\|_X^*)^2\right]}{1 - 2\|V\|_X^*}$$

$$\leq \left(1 - \frac{1 - 2\rho}{1 - \rho} \alpha \right) g_k(X) + \alpha \cdot \frac{\sqrt{\mu/g_{\max}}}{1 - \rho} \cdot g_{\max} + \alpha^2 \cdot \frac{O(d^2 \cdot g_{\max} \cdot \rho^2)}{1 - O(g_{\max}^{1/2} \cdot \rho)}$$

$$\leq \left(1 - \zeta \alpha \right) g_k(X) + \alpha \cdot \frac{\zeta}{2} g_{\max} \qquad \text{for } \alpha = O(\rho^{-1} d^{-2}). \tag{7}$$

It then follows that $g_k(X_+)$ converges geometrically towards $\frac{1}{2}g_{\max}$ in expectation, with a convergence rate $(1 - \zeta\alpha)$ that is independent of the condition number $\kappa$:

$$\mathbb{E}[g_k(X_+) - \frac{1}{2}g_{\max}] \leq \left[(1 - \zeta\alpha)\, g_k(X) + \alpha \cdot \frac{\zeta}{2}g_{\max}\right] - \frac{1}{2}g_{\max} \leq (1 - \zeta\alpha)\left[g_k(X) - \frac{1}{2}g_{\max}\right].$$

The proof of Theorem 2 then follows from standard techniques, by making the two decrement conditions (6) and (7) into supermartingales and applying a standard concentration inequality. We defer the rigorous proof to appendix E.

## 5  Experimental validation

In this section we compare the practical performance of ScaledSGD and SGD for the RMSE loss function in Theorem 2 and two real-world loss functions: the pairwise RMSE loss used to complete Euclidean Distance Matrices (EDM) in wireless communication networks; and the Bayesian Personalized Ranking (BRP) loss used to generate personalized item recommendation in collaborative filtering. In each case, ScaledSGD remains highly efficient since it only updates two or three rows at a time, and the preconditioner $P$ can be computed through low-rank updates, for a per-iteration cost of $O(r^2)$. All of our experiments use random Gaussian initializations and an initial $P = \sigma^2 I$. To be able to accurately measure and report the effects of ill-conditioning on ScaledSGD and SGD, we focus on small-scale synthetic datasets in the first two experiments, for which the ground truth is explicitly known, and where the condition numbers can be finely controlled. In addition, to gauge the scalability of ScaledSGD on huge-scale real-world datasets, in the third experiment, we apply ScaledSGD to generate personalized item recommendation using MovieLens25M dataset [50], for which the underlying item-item matrix has more than 62,000 items and 100 million pairwise samples are used during training. (Due to space constraints, we defer the details on the experimental setup, mathematical formulations, and the actual update equations to Appendix A.) The code for all experiments are available at `https://github.com/Hong-Ming/ScaledSGD`.

**Matrix completion with RMSE loss.**  The problem formulation is discussed in Section 3. Figure 1 plots the error $f(X) = \|XX^T - M\|_F^2$ as the number of epochs increases. As expected, in the well-conditioned case, both ScaledSGD and SGD converges to machine error at roughly the same linear rate. However, in the ill-conditioned case, SGD slows down significantly while ScaledSGD converges at almost exactly the same rate as in the well-conditioned case.

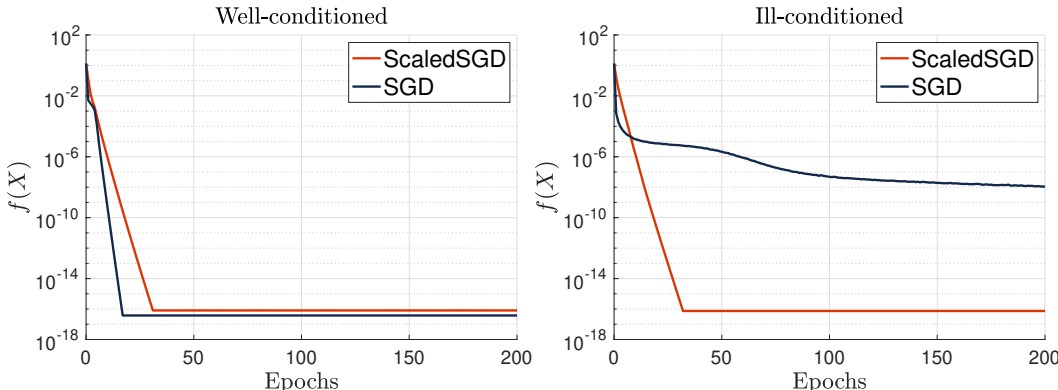

Figure 1: **Matrix Completion with RMSE loss.** We compare the convergence rate of ScaledSGD and SGD for a well-conditioned and ill-conditioned ground truth matrix of size $30 \times 30$ and rank 3. (**Left**) Well-conditoned $M$, $\kappa(M) = 1$. Step-size $\alpha = 0.3$. Both ScaledSGD and SGD converges quickly to the ground truth. (**Right**) Ill-conditoned $M$, $\kappa(M) = 10^4$. Step-size $\alpha = 0.3$. SGD stagnates while ScaledSGD retains the same convergence rate as the well-conditioned case.

**Euclidean distance matrix (EDM) completion.**  The Euclidean distance matrix (EDM) is a matrix of pairwise distance between $d$ points in Euclidean space [51]. In applications such as wireless sensor networks, estimation of unknown distances, i.e., completing the EDM is often required. We

emphasize that this loss function is a pairwise loss, meaning that each measurement indexes multiple elements of the ground truth matrix.

To demonstrate the efficacy of ScaledSGD, we conduct two experiments where $D$ is well-conditioned and ill-conditioned respectively: **Experiment 1.** We uniformly sample 30 points in a cube center at origin with side length 2, and use them to compute the ground truth EDM $D$. In this case, each row $x_i \in \mathbb{R}^3$ corresponds to the coordinates of the $i$-th sample. The corresponding matrix $X \in \mathbb{R}^{30 \times 3}$ is well-conditioned because of the uniform sampling. **Experiment 2.** The ground truth EDM is generated with 25 samples lie in the same cube in experiment 1, and 5 samples lie far away from the the cube. These five outliers make the corresponding $X$ become ill-conditioned.

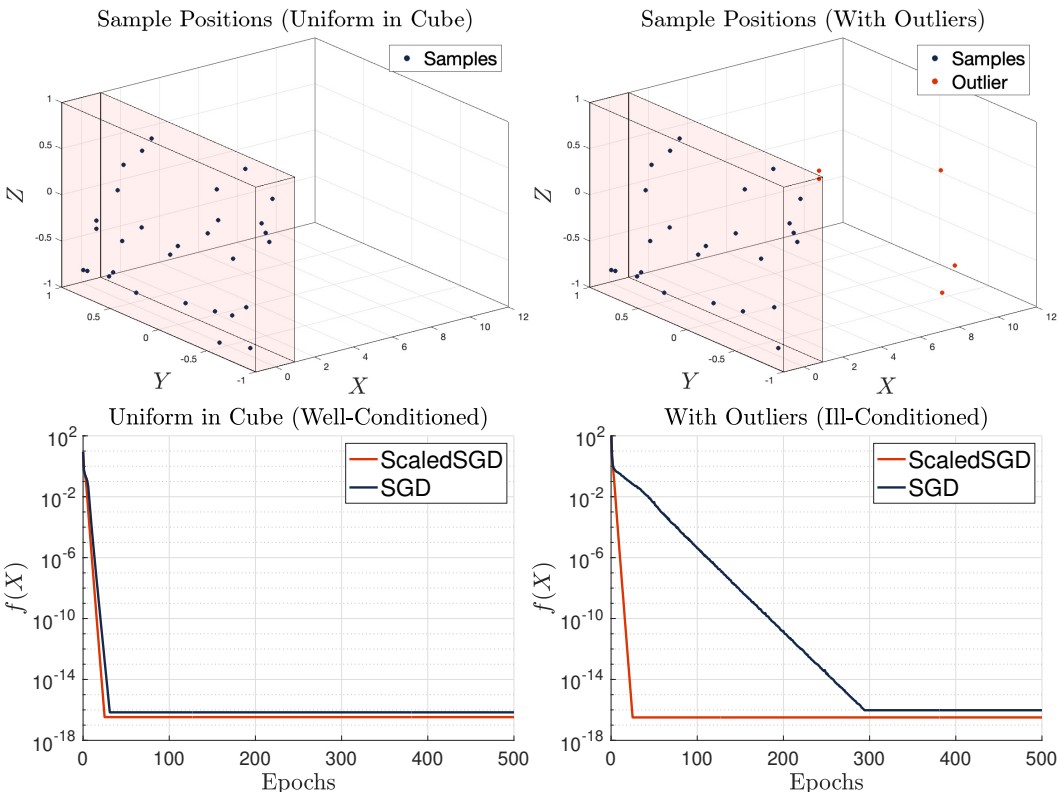

Figure 2: **Euclidean distance matrix (EDM) completion.** We compare the convergence rate of ScaledSGD and SGD for EDM completion for two set of samples. (**Upper right**) 30 samples are uniformly distributed in the pink cube center at origin. (**Upper left**) 25 samples (in blue) are uniformly distributed in the cube, 5 outlier samples (in orange) are outside of the cube. (**Lower left**) Sample uniformly in cube. (**Lower right**) Sample with outliers.

**Item-item collaborative filtering (CF).** In the task of item-item collaborative filtering (CF), the ground truth $M$ is a $d \times d$ matrix where $d$ is the number of items we wish to rank and the $i, j$-th of $M$ is a similarity measure between the items. Our goal is to learn a low-rank matrix that preserves the ranking of similarity between the items. For instance, given a pairwise sample $(i, j, k)$, if item $i$ is more similar to item $j$ than item $k$, then $M_{ij} > M_{ik}$. We want to learn a low-rank matrix that also has this property, i.e., the $i, j$-th entry is greater than the $i, k$-th entry.

To gauge the scalability of ScaledSGD on a huge-scale real-world dataset, we perform simulation on item-item collaborative filtering using a $62,000 \times 62,000$ item-item matrix $M$ obtained from MovieLens25M dataset. The CF model is trained using Bayesian Personalized Ranking (BRP) loss [1] on a training set, which consists of 100 million pairwise samples in $M$. The performance of CF model is evaluated using Area Under the ROC Curve (AUC) score [1] on a test set, which consists of 10 million pairwise samples in $M$. The BPR loss is a widely used loss function in the context of collaborative filtering for the task of personalized recommendation, and the AUC score is

a popular evaluation metric to measure the accuracy of the recommendation system. We defer the detail definition of BPR loss and AUC score to Appendix A.4.

Figure 3 plots the training BPR loss and testing AUC score within the first epoch (filled with red) and the second epoch (filled with blue). In order to measure the efficacy of ScaledSGD, we compare its testing AUC score against a standard baseline called the NP-Maximum [1], which is the best possible AUC score by *non-personalized* ranking methods. For a rigorous definition, see Appendix A.4.

We emphasize two important points in the Figure 3. First, the percentage of training samples needed for ScaledSGD to achieve the same testing AUC scores as NP-Maximum is roughly 4 times smaller than SGD. Though both ScaledSGD and SGD are able to achieve higher AUC score than NP-Maximum before finishing the first epoch, ScaledSGD achieve the same AUC score as NP-Maximum after training on 11% of training samples while SGD requires 46% of them. We note that in this experiment, the size of the training set is 100 million, this means that SGD would require 35 million more iterations than ScaledSGD before it can reach NP-Maximum.

Second, the percentage of training samples needed for ScaledSGD to converge after the first epoch is roughly 5 times smaller than SGD. Given that both ScaledSGD and SGD converge to AUC score at around 0.9 within the second epoch (area filled with blue), we indicate the percentage of training samples when both algorithms reach 0.9 AUC score in Figure 3. As expected, ScaledSGD is able to converge using fewer samples than SGD, with only 16% of training samples. SGD, on the other hand, requires 81% training samples.

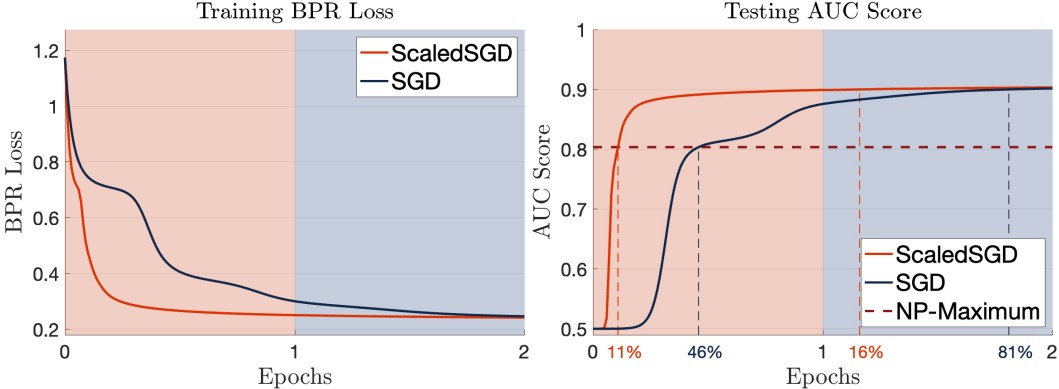

Figure 3: **Huge-scale item-item collaborative filtering.** (MovieLens25M dataset with $|\Omega_{\text{train}}| = 100$ million and $|\Omega_{\text{test}}| = 10$ million pairwise measurements). We compare the training BPR loss and testing AUC score of ScaledSGD and SGD. (**Left**) Training BPR loss on the training set $\Omega_{\text{train}}$. (**Right**) Testing AUC score on the test set $\Omega_{\text{test}}$.

## 6   Conclusions

We propose an algorithm called ScaledSGD for huge scale online matrix completion. For the non-convex approach to solving matrix completion, ill-conditioning in the ground truth causes SGD to slow down significantly. ScaledSGD preserves all the favorable qualities of SGD while making it immune to ill-conditioning. For the RMSE loss, we prove that with an initial point close to the ground truth, ScaledSGD converges to an $\epsilon$-accurate solution in $O(\log(1/\epsilon))$ iterations, independent of the condition number $\kappa$. We also run numerical experiments on a wide range of other loss functions commonly used in applications such as collaborative filtering, distance matrix recovery, etc. We find that ScaledSGD achieves similar acceleration on these losses, which means that it is widely applicable to many real problems. It remains future work to provide rigorous justification for these observations.

## Acknowledgments

The authors thank Salar Fattahi for helpful discussions and feedback on an earlier draft. Financial support for this work was provided in part by the NSF CAREER Award ECCS-2047462 and in part by C3.ai Inc. and the Microsoft Corporation via the C3.ai Digital Transformation Institute.

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
