## A  Supplemental Details on Experiments in Main Paper

### A.1  Experimental setup and datasets used

**Simulation environment.**   We implement both ScaledSGD and SGD in MATLAB (version R2021a). All simulations in this paper are performed on a computer with Apple silicon M1 pro chip with 10-core CPU, 16-core GPU, and 32GB of RAM.

**Datasets.**   The datasets we use for the experiments in the main paper are described below.

- **Matrix completion with RMSE loss:** In the simulation result shown in Figure 1, we synthetically generate both the well-conditioned and ill-conditioned ground truth matrix $M$. We fix both $M$ to be a rank-3 matrix of size $30 \times 30$. To generate $M$, we sample a random orthonormal matrix $U \in \mathbb{R}^{30 \times 3}$ and set $M = USU^T$. For well-conditioned case, we set $S = \text{diag}(2, 2, 2)$, thus $M$ is perfectly conditioned with $\kappa = 1$. For ill-conditioned case, we let $S = \text{diag}(10, 10^{-1}, 10^{-3})$, so that $M$ is ill-conditioned with $\kappa = 10^4$.

- **Euclidean distance matrix completion:** In this simulation shown in Figure 2, the ground truth Euclidean distance matrix $D$ for experiments 1 and 2 are generated with respect to their sample matrix $X$ as $D_{ij} = \|x_i - x_j\|_2^2$. For the sample points in **Experiment 1**, we randomly sample (without replacement) 30 points in 3-dimensional cube centered at origin with side length 2, and the corresponding sample matrix $X$ has conditioned number $\kappa = 1.3908$. For the sample points in **Experiment 2**, we take the first 5 sample points in experiment 1 and perturb its x-coordinate by 10, and keep the rest of the 25 samples intact. The corresponding sample matrix $X$ has conditioned number $\kappa = 8.0828$.

- **Item-item collaborative filtering:** In this simulation shown in Figure 3, we use the Movie-Lens25M dataset [50], which is a standard benchmark for algorithms for recommendation systems.[2] This dataset consists of 25 million ratings over 62,000 movies by 162,000 users, the ratings are stored in an user-item matrix $G$ whose $(i, j)$-th entry is the rating that the $i$-th user gives the $j$-th movie. The rating is from 1 to 5, where a higher score indicates a stronger preference. If the $i, j$-th entry is 0, then no rating is given. For the simulation of item-item collaborative filtering, the $(i, j)$-th entry of the ground truth item-item matrix $M$ is the *similarity score* between the item $i$ and $j$, which can be computed by measuring cosine similarity between the $i$-th and $j$-th column of $G$.

**Hyperparameter and initialization.**   We start ScaledSGD and SGD at the same initial point in each simulation. The initial points for each simulation are drawn from the standard Gaussian distribution.

- **Matrix completion with RMSE loss:** The step-size for both ScaledSGD and SGD are set to be $\alpha = 0.3$. The search rank for both ScaledSGD and SGD are set to be $r = 3$.

- **Euclidean distance matrix completion:** Since SGD is only stable for small step-size in EDM completion problem, while ScaledSGD can tolerance larger step-sizes, we pick the

---

[2]The MovieLens25M dataset is accessible at https://grouplens.org/datasets/movielens/25m/

largest possible step-size for ScaledSGD and SGD in both experiments. **Experiment 1**: step-size for ScaledSGD $\alpha = 0.2$, step-size for SGD $\alpha = 0.02$. **Experiment 2**: step-size for ScaledSGD $\alpha = 0.2$, step-size for SGD $\alpha = 0.002$. The search rank for both ScaledSGD and SGD are set to be $r = 3$.

- **Item-item collaborative filtering:** The step-sizes for this experiment are set as follows: we first pick a small step-size and train the CF model over a sufficient number of epochs, this allows us to estimate the best achievable AUC score; we then set the step-sizes for both ScaledSGD and SGD to the largest possible step-size for which ScaledSGD and SGD is able to converge to the best achievable AUC score, respectively. The step-size for ScaledSGD $\alpha = 1,000$, step-size for SGD $\alpha = 0.01$. The search rank for both ScaledSGD and SGD are set to be $r = 3$.

## A.2 Matrix completion with RMSE loss

We now turn to the practical aspects of implementing ScaledSGD for RMSE loss function. In practical setting, suppose we are given a set $\Omega = \{(i, j)\}$ that contains indices for which we know the value of $M_{ij}$, our goal is to recover the missing elements in $M$ by solving the following nonconvex optimization

$$\min_{X \in \mathbb{R}^{d \times r}} f(X) = \frac{1}{2|\Omega|} \sum_{(i,j) \in \Omega} \left(x_i^T x_j - M_{ij}\right)^2.$$

The gradient of $f(X)$ is

$$\nabla f(X) = \frac{1}{|\Omega|} \sum_{(i,j) \in \Omega} \left(x_i^T x_j - M_{ij}\right) \left(e_i e_j^T + e_j e_i^T\right) X.$$

**ScaledSGD update equations for RMSE loss.** Each iteration of ScaledSGD samples one element $(i, j) \in \Omega$ uniformly. The resulting iteration updates only two rows of $X$

$$x_{i,+} = x_i - \alpha \cdot \left(x_i^T x_j - M_{ij}\right) P x_j, \quad x_{j,+} = x_j - \alpha \cdot \left(x_i^T x_j - M_{ij}\right) P x_i.$$

The update on $P$ is low-rank

$$P_+ = (P^{-1} + x_{i,+} x_{i,+}^T + x_{j,+} x_{j,+}^T - x_i x_i^T - x_j x_j^T)^{-1},$$

and can be computed by calling four times of rank-1 Sherman–Morrison–Woodbury (SMW) update formula in $O(r^2)$ time

$$(P^{-1} + uu^T)^{-1} = P - \frac{Puu^T P}{1 + u^T P u}, \qquad (P^{-1} - uu^T)^{-1} = P + \frac{Puu^T P}{1 - u^T P u}. \qquad (8)$$

In practice, this low-rank update can be "pushed" onto a centralized storage of the preconditioner $P$. Heuristically, independent copies of $P$ can be maintained by separate, distributed workers, and a centralized dispatcher can later merge the updates to $P$ by simply adding the cumulative low-rank updates onto the existing centralized copy.

## A.3 Euclidean distance matrix (EDM) completion

Suppose that we have $d$ points $x_1, \ldots, x_d \in \mathbb{R}^r$ in $r$ dimensional space, the Euclidean distance matrix $D \in \mathbb{R}^{d \times d}$ is a matrix of pairwise squared distance between $d$ points in Euclidean space, namely, $D_{ij} = \|x_i - x_j\|^2$. Many applications, such as wireless sensor networks, communication and machine learning, require Euclidean distance matrix to provide necessary services. However, in practical scenario, entries in $D$ that correspond to points far apart are often missing due to high uncertainty or equipment limitations in distance measurement. The task of Euclidean distance matrix completion is to recover the missing entries in $D$ from a set of available measurement, and this problem can be formulated as a rank $r$ matrix completion problem with respect to pairwise square loss function. Specifically, let $X \in \mathbb{R}^{d \times r}$ be a matrix containing $x_1, \ldots, x_d$ in its row and let $M = XX^T$ be the Grammian of $X$. Each entry of $D$ can be written in terms of three entries in $M$

$$D_{ij} = \|x_i - x_j\|^2 = x_i^T x_i - 2x_i^T x_j + x_j^T x_j = M_{ii} - 2M_{ij} + M_{jj}.$$

Hence, given a set of sample $\Omega = \{(i,j)\}$ in $D$, the pairwise square loss function for EDM completion reads

$$\min_{X \in \mathbb{R}^{d \times r}} f(X) = \frac{1}{4|\Omega|} \sum_{(i,j) \in \Omega} \left( x_i^T x_i - 2x_i^T x_j + x_j^T x_j - D_{ij} \right)^2.$$

The gradient of $f(X)$ is

$$\nabla f(X) = \frac{1}{|\Omega|} \sum_{(i,j) \in \Omega} \left( x_i^T x_i - 2x_i^T x_j + x_j^T x_j - D_{ij} \right) \left[ \left( e_i e_i^T + e_j e_j^T \right) - \left( e_i e_j^T + e_j e_i^T \right) \right] X.$$

**ScaledSGD update equations for EDM completion.** Each iteration of ScaledSGD samples one element $(i,j) \in \Omega$ uniformly. The resulting iteration updates only two rows of $X$

$$x_{i,+} = x_i - \alpha \cdot \left( x_i^T x_i - 2x_i^T x_j + x_j^T x_j - D_{ij} \right) P(x_i - x_j),$$
$$x_{j,+} = x_j - \alpha \cdot \left( x_i^T x_i - 2x_i^T x_j + x_j^T x_j - D_{ij} \right) P(x_j - x_i)$$

Similarly, the update on $P$ is low-rank and can be computed by calling four times of equation (8).

## A.4 Item-item collaborative filtering (CF)

In the task of item-item collaborative filtering, the ground truth $M$ is an $d \times d$ matrix where $d$ is the number of items we wish to rank and the $i,j$-th of $M$ is a similarity measure between the items. Our goal is to learn a low-rank matrix that preserves the ranking of similarity between the items. For instance, suppose that item $i$ is more similar to item $j$ than item $k$, then $M_{ij} > M_{ik}$, we want to learn a low-rank matrix $XX^T$ that also has this property, i.e., $x_i^T x_j \geq x_i^T x_k$ where $x_i$ is the $i$-th row of $X$.

**Similarity score.** An important building block of item-item recommendation systems is the so-called item-item similarity matrix [52–54], which we denote by $M$. The $i,j$-th entry of this matrix is the pairwise *similarity scores* of items $i$ and $j$. There are various measures of similarity. In our experiments we adopt a common similarity measure known as cosine similarity [53]. As a result, the item-item matrix can be computed from the user-item matrix. In particular, let $g_i$, $g_j$ denote the $i$-th and $j$-th columns of the user-item matrix $G$, corresponding to the ratings given by all users to the $i$-th and $j$-th items. Then the $(i,j)$-th element of the item-item matrix $M$ is set to

$$M_{ij} = g_i^T g_j / (\|g_i\| \|g_j\|).$$

In general, the item-item matrix computed this way will be very sparse and not capable of generating good recommendations. Our goal is to complete the missing entries of this matrix, assuming that that $M$ is low-rank. As we will see, we can formulate this completion problem as an optimization problem over the set of rank-$r$ matrices.

**Pairwise entropy loss (BPR loss).** The Bayesian Personalized Ranking (BRP) loss [1] is a widely used loss function in the context of collaborative filtering. For the task of predicting a personalized ranking of a set of items (videos, products, etc.), BRP loss often outperforms RMSE loss because it is directly optimized for ranking; most collaborative filtering models that use RMSE loss are essentially scoring each individual item based on user implicit feedbacks, in applications that only positive feedbacks are available, the models will not be able to learn to distinguish between negative feedbacks and missing entries.

The BPR loss in this context can be defined as follows. Let $\Omega = \{(i,j,k)\}$ denote a set of indices for which we observe the ranking of similarity between items $i, j, k$. Our observations are of the form $Y_{ijk} = 1$ if $M_{ij} > M_{ik}$ and $Y_{ijk} = 0$ otherwise. In other words, $Y_{ijk} = 1$ if item $i$ is more similar to item $j$ than to item $k$. We form a candidate matrix of the form $XX^T$, where $X \in \mathbb{R}^{d \times r}$. Our hope is that $XX^T$ preserves the ranking between the items. The BPR loss function is designed to enforce this property.

Let $x_i$ denote the $i$-th row of $X$ and set $z_{ijk} = (XX^T)_{ij} - (XX^T)_{ik} = x_i^T(x_j - x_k)$. The BPR loss attempts to preserve the ranking of samples in each row of $M$ by minimizing the logistic loss with respect to $(Y_{ijk}, \sigma(z_{ijk}))$, where $\sigma(\cdot)$ is the sigmoid function:

$$\min_{X \in \mathbb{R}^{n \times r}} f(X) = \frac{1}{|\Omega|} \sum_{(i,j,k) \in \Omega} -Y_{ijk} \log \left( \sigma(z_{ijk}) \right) - (1 - Y_{ijk}) \log \left( 1 - \sigma(z_{ijk}) \right).$$

Then the gradient of $f(X)$ is

$$\nabla f(X) = \frac{1}{|\Omega|} \sum_{(i,j,k)\in\Omega} (\sigma(z_{ijk}) - Y_{ijk}) \left[ \left( e_i e_j^T + e_j e_i^T \right) - \left( e_i e_k^T + e_k e_i^T \right) \right] X.$$

**ScaledSGD update equations for BPR loss.** Similarly to the previous section, each iteration of ScaledSGD samples one element $(i,j,k) \in \Omega$ uniformly. The resulting iteration updates only three rows of $X$, as in

$$x_{i,+} = x_i - \alpha \cdot (\sigma(z_{ijk}) - Y_{ijk}) P(x_j - x_k), \quad x_{j,+} = x_j - \alpha \cdot (\sigma(z_{ijk}) - Y_{ijk}) P x_i,$$
$$x_{k,+} = x_k + \alpha \cdot (\sigma(z_{ijk}) - Y_{ijk}) P x_i$$

Similar to before, the preconditioner $P$ can be updated via six call of equation (8) in $O(r^2)$ time.

**The AUC score.** The AUC score [1] is a popular evaluation metric for recommendation system. Roughly speaking, the AUC score of a candidate matrix $XX^T$ is the percentage of ranking of the entries of $M$ that is preserved by $XX^T$. Specifically, for each sample $(i,j,k) \in \Omega$, we define a indicator variable $\delta_{ijk}$ as

$$\delta_{ijk} = \begin{cases} 1 & \text{if } z_{ijk} > 0 \text{ and } Y_{ijk} = 1 \\ 1 & \text{if } z_{ijk} \le 0 \text{ and } Y_{ijk} = 0 \\ 0 & \text{otherwise,} \end{cases}$$

where we recall that $Y_{ijk}$ is our observation and $z_{ijk} = (XX^T)_{ij} - (XX^T)_{ik}$. In other words, $\delta_{ijk} = 1$ only if the ranking between $M_{ij}$ and $M_{ik}$ is preserved by $(XX^T)_{ij}$ and $(XX^T)_{ik}$. The AUC score is then defined as the ratio

$$\text{AUC} = \frac{1}{|\Omega|} \sum_{(i,j,k)\in\Omega} \delta_{ijk}.$$

Thus, a higher AUC score indicates that the candidate matrix $XX^T$ perserves a larger percentage of the pairwise comparisons in $|\Omega|$.

**Training a CF model for Figure 3.** We precompute a dataset $\Omega$ of 110 million item-item pairwise comparisons using the user-item ratings from the MovieLens25M dataset, and then run ScaledSGD and SGD over 2 epochs on this dataset. Let $d = 62,000$ denote the number of items and let $\Omega \subseteq [d]^3$ denote the set of observations, i.e., the set of entries $(i,j,k)$ where we observe $Y_{ijk} = 1$ if $M_{ij} > M_{ik}$ and $Y_{ijk} = 0$ if $M_{ij} < M_{ik}$. We construct $\Omega$ by sampling 110 million pairwise measurements that have either $M_{ij} > M_{ik}$ or $M_{ij} < M_{ik}$ uniformly at random without replacement from $(i,j,k) \sim [d]^3$. We do this because the item-item matrix $M$ remains highly sparse, and there are many pairs of $(i,j)$ and $(i,k)$ for which $M_{ij} = M_{ik} = 0$.

To ensure independence between training and testing, we divide the set $\Omega$ into two disjoint sets $\Omega_{\text{train}}$ and $\Omega_{\text{test}}$. The first set $\Omega_{\text{train}}$ consists of 100 million of all observations, which we use to fit our model. The second set $\Omega_{\text{test}}$ consists of 10 million samples for which we use to calculate the AUC score of our model on new data.

**Upper bounds on the non-personalized ranking AUC score (NP-Maximum).** As opposed to personalized ranking methods, non-personalized ranking methods generate the same ranking for every pair of item $j$ and $k$, independent of item $i$. In the context of item-item collaborative filtering, the non-personalized ranking method can be defined as follows. Given a set of pairwise comparisons $\Omega = (\{i,j,k\})$ and observations $Y_{ijk}$, we optimized the ranking between item $j$ and $k$ on a candidate vector $x$, where $x \in \mathbb{R}^d$.

Let $x_i$ denote the $i$-th entry of $x$, the non-personalized ranking method attempts to preserve the ranking between the $x_j$ and $x_k$ by minimizing the logistic loss with respect to $(Y_{ijk}, \sigma(x_j - x_k))$ where $\sigma(\cdot)$ is the sigmoid function:

$$\min_{x\in\mathbb{R}^r} f(x) = \frac{1}{|\Omega|} \sum_{(i,j,k)\in\Omega} -Y_{ijk} \log(\sigma(x_j - x_k)) - (1 - Y_{ijk}) \log(1 - \sigma(x_j - x_k)).$$

The gradient of $f(x)$ is

$$\nabla f(x) = \frac{1}{|\Omega|} \sum_{(i,j,k)\in\Omega} (\sigma(x_j - x_k) - Y_{ijk}) \left[(e_j^T - e_k^T)\right] x,$$

and the SGD update equations for $x_j$ and $x_k$ are

$$x_{j,+} = x_j - \alpha \cdot (\sigma(x_j - x_k) - Y_{ijk})\, x_j, \quad x_{k,+} = x_k + \alpha \cdot (\sigma(x_j - x_k) - Y_{ijk})\, x_k.$$

Notice that non-personalized ranking method is not a matrix completion problem, the regular SGD is used to minimized $f(x)$. To find the upper bound on the non-personalized ranking AUC score, we directly optimize the non-personalized ranking on the test set $\Omega_{\text{test}}$, and evaluated the corresponding AUC score on $\Omega_{\text{test}}$. Since we perform both training and evaluation on $\Omega_{\text{test}}$, this corresponding AUC score is the upper bound on the best achievable AUC score on $\Omega_{\text{test}}$.

# B  Additional Experiments on pointwise cross-entropy loss

This problem is also known as 1-bit matrix completion [55]. Here our goal is to recover a rank-$r$ matrix $M$ through *binary* measurements. Specifically, we are allowed to take independent measurements on every entry $M_{ij}$, which we denote by $Y_{ij}$. Let $\sigma(\cdot)$ denote the sigmoid function, then $Y_{ij} = 1$ with probability $\sigma(M_{ij})$ and $Y_{ij} = 0$ otherwise. After a number of measurements are taken on each entry in the set $\Omega$, let $y_{ij}$ denote the percentage of measurements on the $i,j$-th entry that is equal to 1. The plan is to find the maximum likelihood estimator for $M$ by minimizing a cross-entropy loss defined as follow

$$\min_{X\in\mathbb{R}^{d\times r}} f(X) = \frac{1}{|\Omega|} \sum_{(i,j)\in\Omega} -y_{ij} \log\left(\sigma(x_i^T x_j)\right) - (1 - y_{ij}) \log\left(1 - \sigma(x_i^T x_j)\right).$$

We assume an ideal case where the number of measurements is large enough so that $y_{ij} = \sigma(M_{ij})$ and the entries are fully observed. The gradient of $f(X)$ is

$$\nabla f(X) = \frac{1}{|\Omega|} \sum_{(i,j)\in\Omega} \left(\sigma(x_i^T x_j) - y_{ij}\right) \left(e_i e_j^T + e_j e_i^T\right) X.$$

**ScaledSGD update equations for pointwise cross-entropy loss.**  Each iteration of ScaledSGD samples one element $(i,j) \in \Omega$ uniformly. The resulting iteration updates only two rows of $X$

$$x_{i,+} = x_i - \alpha \cdot \left(\sigma(x_i^T x_j) - y_{ij}\right) P x_j, \quad x_{j,+} = x_j - \alpha \cdot \left(\sigma(x_i^T x_j) - y_{ij}\right) P x_i.$$

The preconditioner $P$ can be updated by calling four times of equation (8) as in RMSE loss.

**Matrix completion with pointwise cross-entropy loss.**  We apply ScaledSGD to perform matrix completion through minimizing pointwise cross-entropy loss. In this experiment, the well-conditioned and ill-conditioned ground truth matrix $M$ is the same as those in Figure 1, and the process of data generation are described in A.1. The learning rate for both ScaledSGD and SGD are set to be $\alpha = 1$. The search rank for both ScaledSGD and SGD are set to be $r = 3$.

Figure 4 plots the error $f(X) = \|XX^T - M\|_F^2$ against the number of epochs. Observe that the results shown in Figure 4 are almost identical to that of the RMSE loss shown in Figure 1. Ill-conditioning causes SGD to slow down significantly while ScaledSGD is unaffected.

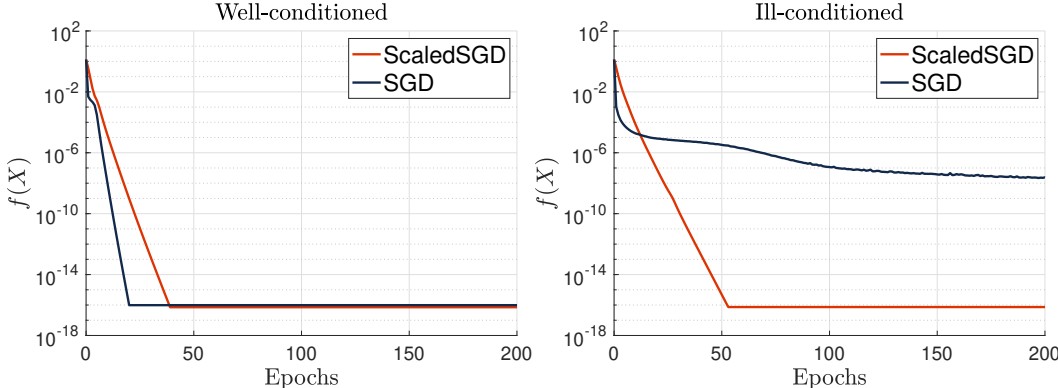

Figure 4: **Matrix Completion with pointwise cross entropy loss.** We compare the convergence rate of ScaledSGD and SGD for a well-conditioned and ill-conditioned ground truth matrix $M$ used in Figure 1. (**Left**) Well-conditoned $M$. (**Right**) Ill-conditoned $M$.

## C   Additional Experiments with Noise

To mimic the real-world datasets, we corrupt each entry of ground truth matrix $M$ by white Gaussian noise. We first generate a noiseless well-conditioned and ill-conditioned matrix $\tilde{M}$ following same procedure as the one described in A.1. For well-conditioned case, we set the singular value as $S = \text{diag}(10, 10, 10)$. For the ill-conditioned case, we set $S = \text{diag}(10, 10^{-1}, 10^{-3})$. To obtain a noisy ground truth, we generate a matrix of white Gaussian noise $W$ corresponding to a fixed signal to noise ratio (SNR), which is defined as $\text{SNR} = 20\log_{10}(\|\tilde{M}\|_F/\|W\|_F)$. Finally, we set $M = \tilde{M} + W$. For the experiments in this section, we set $\text{SNR} = 15\text{dB}$. For the case of well-conditioned $\tilde{M}$, the resulting $M = \tilde{M} + W$ is full-rank with condition number $\kappa = 310.72$. For the case of ill-conditioned $\tilde{M}$, the resulting $M$ is full-rank with condition number $\kappa = 423.5022$.

**Matrix completion with RMSE loss on noisy datasets.** We plot the convergence rate of ScaledSGD and SGD under the noisy setting in Figure 5. In this experiment, we pick a larger search rank $r = 5$ to accommodate the noisy ground truth. Observe that SGD slows down in both the well-conditioned and ill-conditioned case due to the addition of white Gaussian noise and the larger search rank $r$, while ScaledSGD converge linearly toward the noise floor.

We also plot the noise floor, which can be computed as follows. First we take the eigendecomposition of $M = Q\Lambda Q^T$, where $Q$ is an orthonormal matrix and $\Lambda$ is a diagonal matrix containing the eigenvalues of $M$ sorted in descending order in its diagonal entries. Let $\Lambda'$ be a diagonal matrix such that $\Lambda'_{ii} = \Lambda_{ii}$ if $i \leq r$, and $\Lambda'_{ii} = 0$ otherwise, then the noise floor is defined as the RMSE between $M$ and its best rank-$r$ approximation $M' = Q\Lambda'Q^T$, which is equal to $\frac{1}{2|\Omega|}\sum_{(i,j)\in\Omega}\left(M'_{ij} - M_{ij}\right)^2$.

The step-sizes in the simulation are set to be the largest possible step-sizes for which ScaledSGD and SGD can converge to the noise floor. For ScaleSGD, the step-size is set to be $\alpha = 0.15$. For SGD, the step-size is set to be $\alpha = 0.01$. $\text{SNR} = 20\log_{10}(\|\tilde{M}\|_F/\|W\|_F) = 15\text{dB}$.

**Matrix completion with pointwise cross-entropy loss on noisy datasets.** We plot the convergence rate of ScaledSGD and SGD under the noisy setting in Figure 6. Similar to RMSE loss in noisy setting, SGD show down in both well-conditioned and ill-conditioned case, while ScaledSGD converge linearly toward the noise floor. In this simulation, the search rank is set to be $r = 5$. The step-size are set to be the largest possible step-sizes for which ScaledSGD and SGD can converge to the noise floor. For ScaleSGD, the step-size is set to be $\alpha = 0.15$. For SGD, the step-size is set to be $\alpha = 0.01$. $\text{SNR} = 15\text{dB}$.

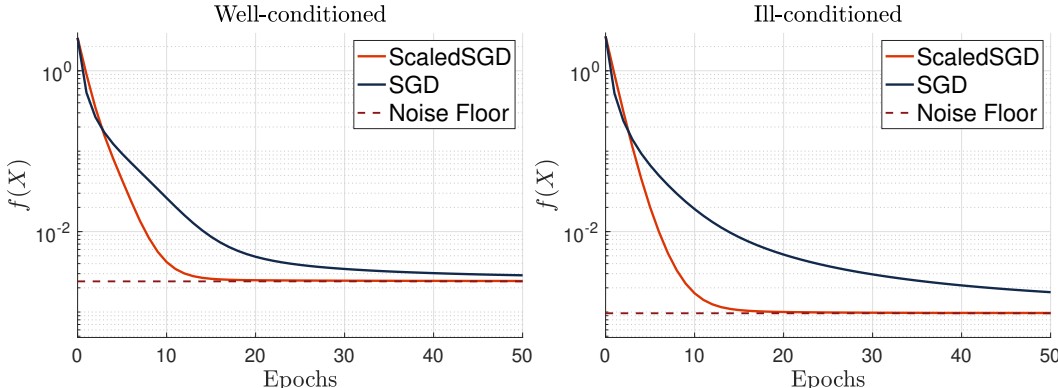

Figure 5: **Matrix Completion with RMSE loss in noisy setting.** We compare the convergence rate of ScaledSGD and SGD for noisy ground truth matrix $M = \tilde{M} + W$ computed with respect to a well-conditioned and ill-conditioned $\tilde{M}$ and white Gaussian noise $W$. (**Left**) Well-conditioned $\tilde{M}$. (**Right**) Ill-conditioned $\tilde{M}$.

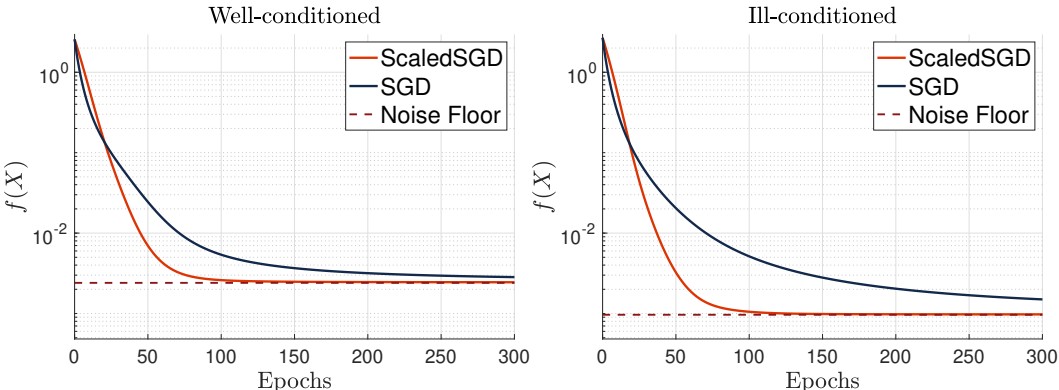

Figure 6: **Matrix Completion with pointwise cross-entropy loss in the noisy setting.** We compare the convergence rate of ScaledSGD and SGD for noisy ground truth matrix $M = \tilde{M} + W$ computed with respect to a well-conditioned and ill-conditioned $\tilde{M}$ and white Gaussian noise $W$. (**Left**) Well-conditioned $\tilde{M}$. (**Right**) Ill-conditioned $\tilde{M}$.

## D  Additional simulation on item-item collaborative filtering

Finally, we perform three additional experiments on item-item collaborative filtering in order to compare the ability of ScaledSGD and SGD to generate good recommendations using matrix factorization.

**Dataset.**  For additional simulations on item-time collaborative filtering, we use the MovieLens-Latest-Small and MovieLens-Latest-Full datasets [50] in order to gauge the performance of our algorithm on different scales. First, we run a small-scale experiment on the MovieLens-Latest-Small dataset that has 100,000 ratings over 9,000 movies by 600 users. Second, we run a medium-scale and a large-scale experiment on the MovieLens-Latest-Full dataset with 27 million total ratings over 58,000 movies by 280,000 users.[3]

**Experimental Setup.**  The process of training a collaborative filtering model is described in A.4. The hyperparameters for the three experiments in this section are described below.

---

[3]Both datasets are accessible at https://grouplens.org/datasets/movielens/latest/

- **MovieLens-Latest-Small dataset:** In the small-scale experiment, we sample $|\Omega_{\text{train}}| = 1$ million and $|\Omega_{\text{test}}| = 100,000$ pairwise observations for training and testing, respectively. We set our search rank to be $r = 3$, so the optimization variable $X$ is of size $9000 \times 3$. Both ScaledSGD and SGD are initialized using a random Gaussian initial point. For ScaledSGD the step-size is $10^3$ and for SGD the step-size is $5 \times 10^{-2}$.

- **MovieLens-Latest-Full dataset:** In the medium-scale experiment, we sample $|\Omega_{\text{train}}| = 10$ million and $|\Omega_{\text{test}}| = 1$ million pairwise observations for training and testing, respectively. In the large-scale experiment, we sample $|\Omega_{\text{train}}| = 30$ million and $|\Omega_{\text{test}}| = 3$ million pairwise observations for training and testing, respectively. In both cases, we set our search rank to be $r = 3$, so the optimization variable $X$ is of size $58000 \times 3$. For ScaledSGD the step-size is $5 \times 10^3$ and for SGD the step-size is $5 \times 10^{-2}$.

**Results.** The results of our experiments for ScaledSGD and SGD are plotted in Figures 7, 8, and 9. In all three cases, ScaledSGD reaches the AUC scores that are greater than NP-Maximum's within the first epoch, while SGD requires more than one epoch to achieve the same AUC score as NP-Maximum's in the small-scale (Figure 7) and medium-scale (Figure 8) setting. In addition, of all three cases, ScaledSGD is able to converge to the asymptote of AUC score within the second epoch, while SGD needs more than 2 epochs to converge to the asymptote in the small-scale (Figure 7) and medium-scale (Figure 8) setting. These results demonstrates that ScaledSGD remain highly efficient across small-scale (Figure 7), medium-scale (Figure 8), large-scale(Figure 9) and huge-scale (Figure 3) settings.

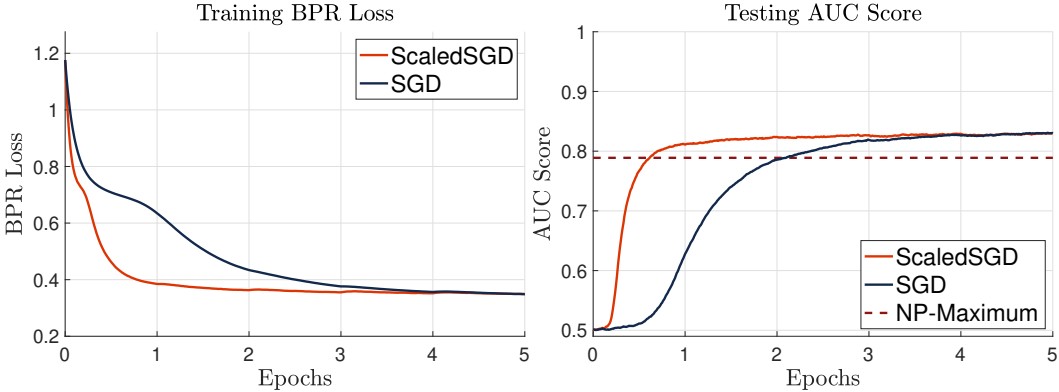

Figure 7: **Small-scale item-item collaborative filtering.** (MovieLens-Latest-Small dataset with $|\Omega_{\text{train}}| = 1$ million and $|\Omega_{\text{test}}| = 100,000$ pairwise measurements). We compare the training BPR loss and testing AUC score of ScaledSGD and SGD. (**Left**) Training BPR loss on the training set $\Omega_{\text{train}}$. (**Right**) Testing AUC score on the test set $\Omega_{\text{test}}$.

# E    Proof of the theoretical results

In this section, we show that, in expectation, the search direction $V = SG(X)(X^T X^{-1})$ makes a geometric decrement to both the function value $f$ and the incoherence $g$. A key idea is to show that the size of the decrement in $f$ is controlled by the coherence $g_{\max} \geq g_k(X)$ of the current iterate, and this motivates the need to decrement $g_k$ in order to keep the iterates incoherent. Our key result is that both decrements are independent of the condition number $\kappa$.

## E.1    Preliminaries

We define the inner product between two matrices as $\langle X, Y \rangle \overset{\text{def}}{=} \text{tr}(X^T Y)$, which induces the Frobenius norm as $\|X\|_F = \sqrt{\langle X, X \rangle}$. The vectorization $\text{vec}(X)$ is the column-stacking operation that turns an $m \times n$ matrix into a length-$mn$ vector; it preserves the matrix inner product $\langle X, Y \rangle = \text{vec}(X)^T \text{vec}(Y)$ and the Frobenius norm $\|\text{vec}(X)\| = \|X\|_F$.

We denote $\lambda_i(M)$ and $\sigma_i(M)$ as the $i$-th eigenvalue and singular value of a symmetric matrix $M = M^T$, ordered from the most positive to the most negative. We will often write $\lambda_{\max}(M)$ and

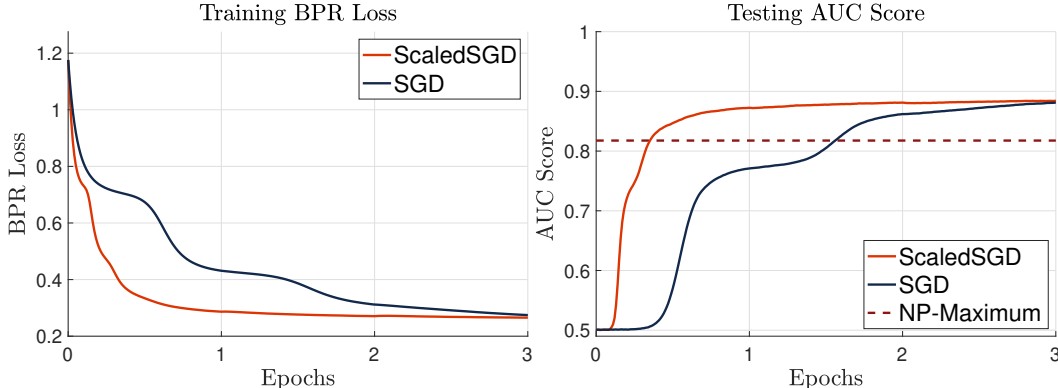

Figure 8: **Medium-scale item-item collaborative filtering.** (MovieLens-Latest-Full dataset with $|\Omega_{\text{train}}| = 10$ million and $|\Omega_{\text{test}}| = 1$ million pairwise measurements). We compare the BPR loss and AUC score of ScaledSGD and SGD (**Left**) Training BPR loss on the training set $\Omega_{\text{train}}$. (**Right**) Testing AUC score on the test set $\Omega_{\text{test}}$.

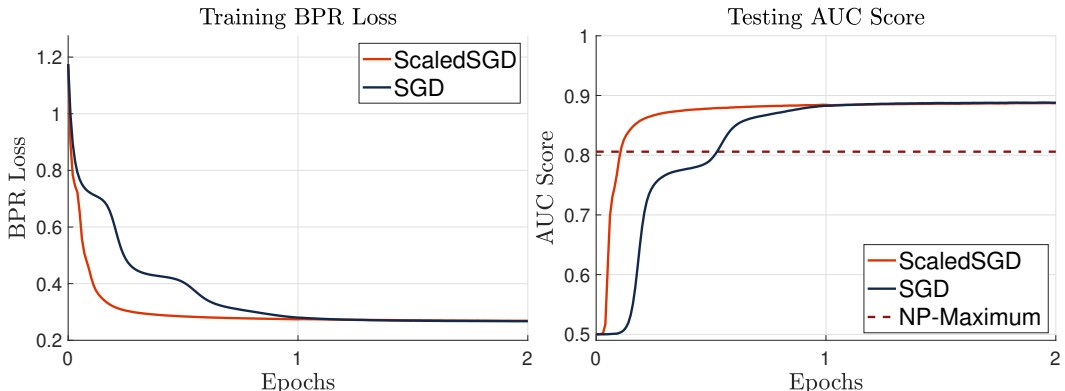

Figure 9: **Large-scale item-item collaborative filtering.** (MovieLens-Latest-Full dataset with $|\Omega_{\text{train}}| = 30$ million and $|\Omega_{\text{test}}| = 3$ million pairwise measurements). We compare the BPR loss and AUC score of ScaledSGD and SGD (**Left**) Training BPR loss on the training set $\Omega_{\text{train}}$. (**Right**) Testing AUC score on the test set $\Omega_{\text{test}}$.

$\lambda_{\min}(M)$ to index the most positive and most negative eigenvalues, and $\sigma_{\max}(M)$ and $\sigma_{\min}(M)$ for the largest and smallest singular values.

Recall for any matrix $V \in \mathbb{R}^{d \times r}$, we define its local norm with respect to $X \in \mathbb{R}^{d \times r}$ as

$$\|V\|_X = \|V(X^T X)^{1/2}\|_F, \qquad \|V\|_X^* = \|V(X^T X)^{-1/2}\|_F.$$

Also recall that we have defined the stochastic gradient operator

$$SG(X) = 2d^2 \cdot (XX^T - ZZ^T)_{i,j} \cdot (e_i e_j^T + e_j e_i^T) X \tag{9}$$

where $(i,j) \sim [d]^2$ is selected uniformly at random. This way SGD is written $X_+ = X - \alpha SG(X)$ and ScaledSGD is written $X_+ = X - \alpha SG(X)(X^T X)^{-1}$ for step-size $\alpha > 0$.

### E.2 Function value convergence

Recall that Theorem 1, due to Jin et al. [10], says that SGD converges to $\epsilon$ accuracy in $O(\kappa^4 \log(1/\epsilon))$ iterations, with a four-orders-of-magnitude dependence on the condition number $\kappa$. By comparison, our main result Theorem 2 says that ScaledSGD converges to $\epsilon$ accuracy in $O(\log(1/\epsilon))$ iterations, completely independence on the condition number $\kappa$. In this section, we explain how the first two factors of $\kappa$ are eliminated, by considering the full-batch counterparts of these two algorithms.

First, consider full-batch gradient descent on the function $f(X) = \|XX^T - ZZ^T\|_F^2$. It follows from the local Lipschitz continuity of $f$ that

$$f(X_{t+1}) \leq f(X_t) - \alpha\langle\nabla f(X_t), \nabla f(X_t)\rangle + \alpha \cdot (L/2) \cdot \|\nabla f(X_t)\|_F^2$$
$$= f(X_t) - \underbrace{\alpha\|\nabla f(X_t)\|_F^2}_{\text{linear progress}} + \alpha^2 \cdot (L/2) \cdot \underbrace{\|\nabla f(X_t)\|_F^2}_{\text{inverse step-size}} \tag{10}$$

where $X_{t+1} = X_t - \alpha\nabla f(X_t)$. Here, the linear progress term determines the amount of progress that can proportionally be made with a sufficiently small step-size $\alpha$, whereas the inverse step-size term basically controls how large the step-size can be. In the case of full-batch gradient descent, it is long known that an $X_t$ that is sufficiently close to $Z$ will satisfy the following

$$8\lambda_{\min}(Z^T Z) \cdot f(X_t) \leq \|\nabla f(X_t)\|_F^2 \leq 16\lambda_{\max}(Z^T Z) \cdot f(X_t)$$

and therefore, taking $\lambda_{\max}(Z^T Z) = 1$ and $\lambda_{\min}(Z^T Z) = \kappa^{-1}$ where $\kappa$ is the condition number, we have linear convergence

$$f(X_{t+1}) \leq \left(1 - \alpha \cdot 8\kappa^{-1} + \alpha^2 \cdot 8L\right) f(X_t) \leq \left(1 - \alpha \cdot 4\kappa^{-1}\right) f(X_t)$$

for step-sizes of $\alpha \leq 2/(\kappa L)$. Therefore, it follows from this analysis that full-batch gradient descent takes $T = O(\kappa^2 \log(1/\epsilon))$ iterations to converge to $\epsilon$-accuracy. In this iteration count, one factor of $\kappa$ arises from the linear progress term, which shrinks as $O(\kappa^{-1})$ as $\kappa$ grows large. The second factor of $\kappa$ arises because the inverse step-size term is a factor of $\kappa$ larger than the linear progress term, which restricts the maximum step-size to be no more than $O(\kappa^{-1})$.

The following lemma, restated from the main text, shows that an analogous analysis for full-batch ScaledGD proves an iteration count of $T = O(\log(1/\epsilon))$ with no dependence on the condition number $\kappa$. In fact, it proves that full-batch ScaledGD converges like full-batch gradient descent with a perfect condition number $\kappa = 1$.

**Lemma 5** (Function descent, Lemma 3 restated). *Let $X, Z \in \mathbb{R}^{n \times r}$ satisfy $\|XX^T - ZZ^T\|_F \leq \rho\lambda_{\min}(Z^T Z)$ where $\rho < 1/2$. Then, the function $f(X) = \|XX^T - ZZ^T\|_F^2$ satisfies*

$$|f(X + V) - f(X) - \langle\nabla f(X), V\rangle| \leq \frac{L_X}{2} \cdot \|V\|_X^2, \tag{11}$$
$$13 \cdot f(X) \leq (\|\nabla f(X)\|_X^*)^2 \leq 16 \cdot f(X), \tag{12}$$

*for all $\|V\|_X \leq C \cdot \sqrt{f(X)}$ with $L_X = 6 + 8C + 2C^2 = O(1 + C^2)$.*

It follows that the iteration $X_{t+1} = X_t - \alpha\nabla f(X_t)(X_t^T X_t)^{-1}$ yields

$$f(X_{t+1}) \leq f(X_t) - \alpha\langle\nabla f(X_t), \nabla f(X_t)(X_t^T X_t)^{-1}\rangle + \alpha \cdot (L_X/2) \cdot \|\nabla f(X_t)(X_t^T X_t)^{-1}\|_X^2$$
$$= f(X_t) - \alpha(\|\nabla f(X_t)\|_X^*)^2 + \alpha^2 \cdot (L_X/2) \cdot (\|\nabla f(X_t)\|_X^*)^2 \tag{13}$$
$$\leq \left(1 - \alpha \cdot 8 + \alpha^2 \cdot 8L_X\right) f(X_t) \leq \left(1 - \alpha \cdot 4\right) f(X_t) \tag{14}$$

for step-sizes of $\alpha \leq 2/L_X$, where $L_X = 6 + 8(4) + 2(4)^2$. Therefore, we conclude that full-batch ScaledGD takes $T = O(\log(1/\epsilon))$ iterations to converge to $\epsilon$-accuracy, as if the condition number were perfectly $\kappa = 1$.

Note that Lemma 5 has been proved in both Tong et al. [9] and Zhang et al. [5]. For completeness, we give a proof inspired by Zhang et al. [5].

*Proof of Lemma 5.* We prove (11) via a direct expansion of the quadratic

$$\underbrace{\|(X + V)(X + V)^T - ZZ^T\|_F^2}_{f(X+V)} = \underbrace{\|XX^T - ZZ^T\|_F^2}_{f(X)} + \underbrace{2\langle XX^T - ZZ^T, XV^T + VX^T\rangle}_{\langle\nabla f(X), V\rangle}$$
$$+ \underbrace{2\langle XX^T - ZZ^T, VV^T\rangle + \|XV^T + VX^T\|_F^2}_{\frac{1}{2}\langle\nabla^2 f(X)[V], V\rangle} + \underbrace{2\langle VX^T + XV^T, VV^T\rangle}_{\frac{1}{6}\langle\nabla^3 f(X)[V,V], V\rangle} + \underbrace{\|VV^T\|_F^2}_{\frac{1}{24}\langle\nabla^4 f(X)[V,V,V], V\rangle}$$

and it then follows by simple counting that

$$|f(X+V) - f(X) - \langle \nabla f(X), V \rangle| \leq \frac{L_2}{2}\|V\|_X^2 + \frac{L_3}{6}\|V\|_X^3 + \frac{L_4}{24}\|V\|_X^4,$$

$$L_2 = 4 + 2\frac{\|XX^T - ZZ^T\|_F}{\lambda_{\min}(X^TX)}, \quad L_3 = \frac{24}{\lambda_{\min}(X^TX)}, \quad L_4 = \frac{24}{\lambda_{\min}^2(X^TX)}.$$

Now, from Weyl's inequality that

$$\lambda_{\min}(X^TX) = \lambda_r(XX^T) \geq \lambda_r(ZZ^T) - \|XX^T - ZZ^T\|_F \geq (1 - \rho) \cdot \lambda_{\min}(Z^TZ)$$

and therefore $\|XX^T - ZZ^T\|_F/\lambda_{\min}(X^TX) \leq \rho/(1-\rho) \leq 1$ because $\rho < 1/2$. It follows that $L_2 \leq 6$. If $\|V\|_X \leq C \cdot \sqrt{f(X)}$, then $\frac{L_3}{3}\|V\|_X \leq 8C$ and $\frac{L_4}{12}\|V\|_X^2 \leq 2C^2$.

For the upper-bound in (12), we have simply

$$\|\nabla f(X)\|_X^* = 4\|(XX^T - ZZ^T)X(X^TX)^{-1/2}\|_F \leq 4\|XX^T - ZZ^T\|_F.$$

For the lower-bound in (12), we evoke Zhang et al. [5, Lemma 12] with RIP constant $\delta = 0$ and regularization parameter $\eta = 0$ to yield[4]

$$\|\nabla f(X)\|_X^* = \max_{\|Y\|_X=1} 2\langle XY^T + YX^T, XX^T - ZZ^T\rangle = 4\|XX^T - ZZ^T\|_F \cdot \cos\theta$$

in which $\cos\theta$ is defined between $XX^T - ZZ^T$ and the set $\{XY^T + YX^T : Y \in \mathbb{R}^{d\times r}\}$, as in

$$\cos\theta = \max_{Y \in \mathbb{R}^{d\times r}} \frac{\langle XY^T + YX^T, XX^T - ZZ^T\rangle}{\|XY^T + YX^T\|_F \cdot \|XX^T - ZZ^T\|_F}.$$

It follows from Zhang et al. [5, Lemma 13] that

$$\sin\theta = \frac{\|(I - XX^\dagger)(XX^T - ZZ^T)(I - XX^\dagger)\|_F}{\|XX^T - ZZ^T\|} \leq \frac{1}{\sqrt{2}}\frac{\rho}{\sqrt{1-\rho^2}}.$$

Hence, for $\rho < 1/2$, we have

$$\frac{(\|\nabla f(X)\|_X^*)^2}{\|XX - ZZ^T\|_F^2} = 16\cos^2\theta \geq 16\left(1 - \frac{1}{2}\frac{\rho^2}{1-\rho^2}\right) \geq 16\left(1 - \frac{1}{6}\right) = \frac{40}{3} > 13.$$

$\square$

### E.3 Coherence convergence

We now explain that ScaledSGD eliminates the last two factors of $\kappa$ from SGD because it is able to keep its iterates a factor of $\kappa^2$ more incoherent. First, consider regular SGD on the function $f(X) = \|XX^T - ZZ^T\|_F^2$. Conditioning on the current iterate, we have via the local Lipschitz continuity of $f$:

$$\mathbb{E}[f(X_+)] \leq f(X) - \alpha\langle \nabla f(X), \mathbb{E}[SG(X)]\rangle + \alpha \cdot (L/2) \cdot \mathbb{E}[\|SG(X)\|_F^2]$$
$$= f(X) - \underbrace{\alpha\|\nabla f(X)\|_F^2}_{\text{linear progress}} + \alpha^2 \cdot (L/2) \cdot \underbrace{\mathbb{E}[\|SG(X)\|_F^2]}_{\text{inverse step-size}} \quad (15)$$

where $X_+ = X - \alpha SG(X)$. In expectation, the linear progress term of SGD coincides with that of full-batch gradient descent in (10). The inverse step-size term, however, is up to a factor of $d^2$ times larger. To see this, observe that

$$\mathbb{E}[\|SG(X)\|_F^2] = \frac{1}{d^2}\sum_{i=1}^d\sum_{j=1}^d \|2d^2 \cdot (XX^T - ZZ^T)_{i,j} \cdot (e_ie_j^T + e_je_i^T)X\|_F^2$$

$$= 4d^2 \cdot \sum_{i=1}^d\sum_{j=1}^d (XX^T - ZZ^T)_{i,j}^2 \cdot \|(e_ie_j^T + e_je_i^T)X\|_F^2$$

$$= 4d^2 \cdot f(X) \cdot \|(e_ie_j^T + e_je_i^T)X\|_F^2 \leq 16d^2 \cdot f(X) \cdot \max_i \|e_i^TX\|_F^2.$$

---

[4]Here, we correct for a factor-of-two error in Zhang et al. [5, Lemma 12].

In a coarse analysis, we can simply bound $\max_i \|e_i^T X\|_F^2 \le \lambda_{\max}(X^T X) = O(1)$ to yield

$$f(X_+) \le \left(1 - \alpha \cdot 8\kappa^{-1} + 16d^2 \cdot \alpha^2\right) f(X) \le \left(1 - \alpha \cdot 4\kappa^{-1}\right) f(X)$$

for step-sizes of $\alpha \le 4/(\kappa d^2)$. Hence, we conclude that it takes $T = O(\kappa^2 d^2 \log(1/\epsilon))$ iterations to converge to $\epsilon$-accuracy, with an epoch of $d^2$ iterations of SGD essentially recreating a single iteration of full-batch gradient descent. Unfortunately, the matrix is already fully observed after $d^2$ iterations, and so this result is essentially vacuous.

Here, Jin et al. [10] pointed out that the term $h_{\max} = \max_i \|e_i^T X\|_F^2$ measures the *coherence* of the $d \times r$ iterate $X$, and can be as small as $O(1/d)$ for small values of rank $r = O(1)$. Conditioned on the current iterate $X$, they observed that the function $h_i(X) = \|e_i^T X\|_F^2$ converges towards a finite value in expectation

$$\mathbb{E}[h_i(X_+)] \le \left(1 - \alpha \cdot 8\kappa^{-1}\right) h_i(X) + \alpha \cdot 8\sqrt{h_i(X)h_i(Z)} + \alpha^2/2 \cdot \mathbb{E}[\|e_i^T SG(X)\|_F^2]$$
$$\le \left(1 - \alpha \cdot 8\kappa^{-1}\right) h_i(X) + \alpha \cdot 8\sqrt{h_i(X)h_i(Z)} + \alpha^2 \cdot O(d^2 h_{\max}^2).$$

Let us define $\gamma$ as the *ratio* between the coherences of the ground truth $Z$ and the iterate $X$:

$$\gamma = \frac{\max_i \|e_i^T X\|_F^2}{\max_j \|e_j^T Z\|_F^2} = \frac{\max_i h_i(Z)}{\max_j h_j(Z)} \quad \Longleftrightarrow \quad \max_j \|e_j^T Z\|_F^2 \le \gamma^{-1} \cdot h_{\max}.$$

Crucially, we require $\gamma = \kappa^2$ in order for $h_i(X)$ to converge towards $\frac{1}{2} h_{\max}$ in expectation:

$$\mathbb{E}[h_i(X_+) - \frac{1}{2} h_{\max}] \le \left(1 - \alpha \cdot 8\kappa^{-1}\right) h_i(X) + \alpha \cdot 4\gamma^{-1/2} h_{\max} - \frac{1}{2} h_{\max}$$
$$\le \left(1 - \alpha \cdot 8\kappa^{-1}\right) \left[h_i(X) - \left(\frac{1 - \alpha \cdot 8\gamma^{-1/2}}{1 - \alpha \cdot 8\kappa^{-1}}\right) \frac{1}{2} h_{\max}\right].$$

As a consequence, we conclude that, while SGD is able to keep its iterates $X$ incoherent, their actual coherence $h_{\max} = \max_i \|e_i^T X\|_F^2$ is up to a factor of $\kappa^2$ worse than the coherence $\max_j \|e_j^T Z\|_F^2$ of the ground truth $Z$.

Using a standard supermartingale argument, Jin et al. [10] extended the analysis above to prove that if the ground truth $Z$ has coherence $\max_j \|e_j^T Z\|_F^2 = O(1/d)$, then the SGD generates iterates $X$ that have coherence $\max_i \|e_i^T X\|_F^2 \le h_{\max} = O((\kappa^2/d) \log d)$, which is two factors worse in $\kappa$ as expected. Combined, this proves that SGD converges to $\epsilon$ accuracy in $T = O(\kappa^4 dr \log(d/\epsilon))$ iterations with the step-size of $\alpha = O(\kappa^{-1} d^{-1} h_{\max}^{-1})$ and iterate coherence $h_{\max} = O((\kappa^2/d) \log d)$, which is another two factors of $\kappa$ worse than full-batch gradient descent.

The following lemma, restated from the main text, shows that an analogous analysis for ScaledSGD proves that the algorithm maintains iterates $X$ whose coherences have no dependence on $\kappa$. Here, we need to define a different incoherence function $g_i(X) = \|e_i X (X^T X)^{-1/2}\|^2 \equiv (\|e_i X\|_X^*)^2$ in order to "stochastify" our previous analysis for full-batch ScaledGD. Surprisingly, the factors of $(X^T X)^{-1}$ in both the new definition of $g_i(X)$ and the search direction $SG(X)(X^T X)^{-1}$ do not hurt incoherence, but in fact improves it.

**Lemma 6** (Coherence descent, Lemma 4 restated). *Let $X, Z \in \mathbb{R}^{n \times r}$ satisfy $\|XX^T - ZZ^T\|_F \le \rho \lambda_{\min}(Z^T Z)$ where $\rho < 1/2$. Then, the functions $f(X) = \|XX^T - ZZ^T\|_F^2$ and $g_k(X) = e_k^T X (X^T X)^{-1} X^T e_k$ satisfy*

$$|g_k(X + V) - g_k(X) - \langle V, \nabla g_k(X) \rangle| \le \frac{5(\|V\|_X^*)^2}{1 - 2\|V\|_X^*},$$

$$\langle \nabla g_k(X), \nabla f(X)(X^T X)^{-1} \rangle \ge \left[\frac{1 - 2\rho}{1 - \rho} g_k(X) - \frac{1}{1 - \rho} \sqrt{g_k(X) g_k(Z)}\right].$$

Conditioning on $X$, we have for the search direction $V = SG(X)(X^T X)^{-1}$ and $X_+ = X + V$

$$\mathbb{E} g_k(X_+) \le g_k(X) - \alpha \langle \nabla g_k(X), \mathbb{E}[V] \rangle + \alpha^2 \cdot \mathbb{E}\left[\frac{(\|V\|_X^*)^2}{1 - 2\|V\|_X^*}\right]$$

$$\le (1 - \zeta\alpha) g_k(X) + \alpha \cdot \frac{\zeta}{2} g_{\max} \qquad \text{for } \alpha = O(\rho^{-1} d^{-2}) \qquad (16)$$

where $\zeta = \frac{1-2\rho}{1-\rho}$. It then follows that $g_k(X_+)$ converges geometrically towards $\frac{1}{2}g_{\max}$ in expectation, with a convergence rate $(1 - \zeta\alpha)$ that is independent of the condition number $\kappa$:

$$\mathbb{E}\left[g_k(X_+) - \frac{1}{2}g_{\max}\right] \le \left[(1 - \zeta\alpha)\, g_k(X) + \alpha \cdot \frac{\zeta}{2}g_{\max}\right] - \frac{1}{2}g_{\max} \le (1 - \zeta\alpha)\left[g_k(X) - \frac{1}{2}g_{\max}\right].$$

Before we prove Lemma 6, we first need to prove a simple claim.

**Lemma 7** (Change of norm). *The local norm $\|V\|_X^* = \|V(X^TX)^{-1/2}\|$ satisfies*

$$\frac{(\|V\|_X^*)^2}{1 + 2\|Y - X\|_X^* + (\|Y - X\|_X^*)^2} \le (\|V\|_Y^*)^2 \le \frac{(\|V\|_X^*)^2}{1 - 2\|Y - X\|_X^*}.$$

*Proof.* The upper-bound follows because

$$\mathrm{tr}(VP_YV^T) = \mathrm{tr}(VP_X^{1/2}[P_X^{-1/2}P_YP_X^{-1/2}]P_X^{1/2}V^T) \le \mathrm{tr}(VP_XV^T)/\lambda_{\min}[P_X^{1/2}P_Y^{-1}P_X^{1/2}]$$

where $P_Y = (Y^TY)^{-1}$ and $P_X = (X^TX)^{-1}$ and therefore

$$P_X^{1/2}P_Y^{-1}P_X^{1/2} \succeq I + P_X^{1/2}[X^T(Y - X) + (Y - X)^TX]P_X^{1/2}$$

and $\sigma_{\max}[P_X^{1/2}[X^T(Y-X)P_X^{1/2}] \le \|Y - X\|_X$ because $XP_X^{1/2}$ is orthonormal. The lower-bound follows similarly. $\square$

We are ready to prove Lemma 6.

*Proof of Lemma 6.* It follows from the intermediate value version of Taylor's theorem that there exists some $\tilde{X} = X + tV$ with $t \in [0, 1]$ that

$$g_i(X + V) - g_i(X) - \langle \nabla g_i(X), V\rangle = \frac{1}{2}\langle \nabla^2 g_i(\tilde{X})[V], V\rangle.$$

Let $P = (X^TX)^{-1}$ and $U = e_ie_i^T(I - XPX^T)V$ and $G = VPX^Te_ie_i^T$. By direct computation, we have

$$\frac{1}{2}\langle \nabla g_i(X), V\rangle = \langle (I - XPX^T)e_ie_i^TXP, V\rangle = \langle U, XP\rangle = \langle I - XPX^T, G\rangle,$$

$$\frac{1}{2}\langle \nabla^2 g_i(X)[V], V\rangle = \langle UP - XP(U^TX + X^TU)P, V\rangle - \langle (I - XPX^T)(G + G^T)XP, V\rangle,$$

by differentiating $XP$ and $XPX^T$ respectively. A coarse count yields $\frac{1}{2}\langle \nabla^2 g_i(X)[V], V\rangle \le 5(\|V\|_X^*)^2$ and therefore

$$|g_i(X + V) - g_i(X) - \langle \nabla g_i(X), V\rangle| \le 5\|V\|_{X+tV}^2 \le \frac{5(\|V\|_X^*)^2}{1 - 2t\|V\|_X^*} \le \frac{5(\|V\|_X^*)^2}{1 - 2\|V\|_X^*},$$

which is the first claim. Now, observe that the two functions have gradient

$$\nabla g_i(X) = 2[I - X(X^TX)^{-1}X^T]e_ie_i^TX(X^TX)^{-1}, \qquad \nabla f(X) = 4(XX^T - ZZ^T)X.$$

Directly substituting yields

$$\frac{1}{8}\langle \nabla g_i(X), \nabla f(X)(X^TX)^{-1}\rangle = \langle [I - X(X^TX)^{-1}X^T]e_ie_i^TX(X^TX)^{-1}, (XX^T - ZZ^T)X(X^TX)^{-1}\rangle$$

$$= e_i^TX(X^TX)^{-2}X^TZZ^TX(X^TX)^{-1}X^Te_i - e_i^TZZ^TX(X^TX)^{-2}X^Te_i$$

where the second line follows from the fact that

$$\langle [I - X(X^TX)^{-1}X^T]e_ie_i^TX(X^TX)^{-1}, XX^TX(X^TX)^{-1}\rangle = 0.$$

The second claim follows from the following three identities

$$\lambda_{\min}(X^TX) \ge (1 - \rho)\lambda_{\min}(Z^TZ) \tag{17}$$

$$e_i^TZZ^TX(X^TX)^{-2}X^Te_i \le \frac{1}{1-\rho}\|e_i^TX\|_X^* \cdot \|e_i^TZ\|_Z^* \tag{18}$$

$$e_i^TX(X^TX)^{-2}X^TZZ^TX(X^TX)^{-1}X^Te_i \ge \frac{1-2\rho}{1-\rho} \cdot (\|e_i^TX\|_X^*)^2 \tag{19}$$

We have (17) via Weyl's inequality:

$$\lambda_{\min}(X^T X) = \lambda_r(XX^T) = \lambda_r(ZZ^T + XX^T - ZZ^T) \geq \lambda_r(ZZ^T) - \|XX^T - ZZ^T\|_F.$$

We have (18) by rewriting

$$e_i^T ZZ^T X(X^T X)^{-2} X^T e_i = (e_i^T P)(P^T ZZ^T X(X^T X)^{-2} X^T Q)(Q^T e_i)$$
$$\leq \|e_i^T P\| \|ZZ^T X(X^T X)^{-2} X^T\| \|e_i^T Q\|$$

and rewriting $ZZ^T = XX^T - E$ where $E = XX^T - ZZ^T$ and evoking (17) as in

$$\|ZZ^T X(X^T X)^{-2} X^T\| \leq \underbrace{\|XX^T X(X^T X)^{-2} X^T\|}_{=1} + \underbrace{\|E\| \cdot \|X(X^T X)^{-2} X^T\|}_{\leq \rho/(1-\rho)}$$

and noting that $1 + \frac{\rho}{1-\rho} = \frac{1}{1-\rho}$. We have (19) again by substituting $ZZ^T = XX^T - E$

$$e_i^T X(X^T X)^{-2} X^T ZZ^T X(X^T X)^{-1} X^T e_i = e_i^T X(X^T X)^{-1} X^T e_i - e_i^T X(X^T X)^{-2} X^T EX(X^T X)^{-1} X^T e_i$$
$$\geq e_i^T X(X^T X)^{-1} X^T e_i \cdot (1 - \underbrace{\|E\| \cdot \|X(X^T X)^{-2} X^T\|}_{\leq \rho/(1-\rho)})$$

and then noting that $1 - \frac{\rho}{1-\rho} = \frac{1-2\rho}{1-\rho}$. $\qquad\square$

## E.4  Proof of the main result

In the previous two subsections, we showed that when conditioned on the current iterate $X_t$, a single step of ScaledSGD $X_{t+1} = X_t - \alpha SG(X_t)(X_t^T X_t)^{-1}$ is expected to geometrically converge both the loss function $f$ and each of the incoherence functions $g_i$, as in

$$\mathbb{E}[f(X_{t+1})] \leq (1-\alpha)f(X_t), \quad \mathbb{E}[g_i(X_{t+1}) - \tfrac{1}{2}g_{\max}] \leq \left(1 - \frac{1-2\rho}{1-\rho}\alpha\right)\left[g_i(X_t) - \tfrac{1}{2}g_{\max}\right].$$

In this section, we will extend this geometric convergence to $T$ iterations of ScaledSGD. Our key challenge is to verify that the *variances* and *maximum deviations* of the sequences $f(X_0), f(X_1), \ldots, f(X_T)$ and $g_i(X_0), g_i(X_1), \ldots, g_i(X_T)$ have the right dependence on the dimension $d$, the radius $\rho$, the condition number $\kappa$, the maximum coherence $g_{\max}$, and the iteration count $t$, so that $T$ iterations of ScaledSGD with a step-size of $\alpha \leq c/[(g_{\max} + \rho)d^2 \log d]$ results in no more than a multiplicative factor of 2 deviation from expectation. Crucially, we must check that the cumulated deviation over $T$ iterations does not grow with the iteration count $T$, and that the convergence rate is independent of the condition number $\kappa$. We emphasize that the actual approach of our proof via the Azuma–Bernstein inequality is textbook; to facilitate a direct comparison with SGD, we organize this section to closely mirror Jin et al. [10]'s proof of Theorem 1.

Let $f_{\max} = \rho^2 \cdot \lambda_{\min}^2(Z^T Z)$ and $g_{\max} = \frac{16}{(1-2\rho)^2} \max_i g_i(Z)$. Our goal is to show that the following event happens with probability $1 - T/d^{10}$:

$$\mathfrak{E}_t \equiv \left\{ f(X) \leq \left(1 - \frac{\alpha}{2}\right)^t \cdot f_{\max}, \quad \max_i g_i(X_\tau) \leq g_{\max} \quad \text{for all } \tau \leq t \right\}, \tag{20}$$

Equivalently, conditioned on event $\mathfrak{E}_t$, we want to prove that the probability of failure at time $t+1$ is $\delta \equiv 1/d^{10}$. We split this failure event into a probability of $\frac{\delta}{2}$ that the function value clause fails to hold, as in $f(X_{t+1}) > (1 - \alpha/2)^t \cdot f_{\max}$, and a probability of $\frac{\delta}{2d}$ that any one of the $d$ incoherence caluses fails to hold, as in $g_i(X_{t+1}) > g_{\max}$. Then, cumulated over $T$ steps, the total probability of failure would be $T \cdot \delta = T/d^{10}$ as desired.

We begin by setting up a supermartingale on the loss function $f$. Our goal is to show that the variance and the maximum deviation of this supermartingale have the right dependence on $\alpha, d, \rho, \kappa, g_{\max}$, so that a step-size of $\alpha \leq c/(g_{\max}d^2 \log d)$ with a sufficiently small $c > 0$ will keep the cumulative deviations over $T$ iterations within a factor of 2. Note that, by our careful choice of the coherence function $g_i$, the following statement for ScaledSGD match the equivalent statements for SGD with a perfect condition number $\kappa = 1$; see Jin et al. [10, Section B.2].

**Lemma 8** (Function value supermartingale). *Let $f(X) = \|XX^T - ZZ^T\|_F^2$. Define $f_{\max} = \rho^2 \cdot \lambda_{\min}^2(Z^T Z)$ and $g_{\max} = \frac{16}{(1-2\rho)^2} \max_i g_i(Z)$. For a sufficiently small $c > 0$, the following with learning rate $\alpha \leq c/(g_{\max}d^2 \log d)$ is a supermartingale*

$$F_t = (1 - \alpha)^{-t} f(X_t) \cdot 1_{\mathfrak{E}_t},$$

*meaning that $\mathbb{E}[F_{t+1}|X_t, \ldots, X_0] \leq F_t$ holds for all $t \in \{0, 1, 2, \ldots\}$. Moreover, there exist sufficiently large constants $C_{\mathrm{dev}}, C_{\mathrm{var}} > 0$ such that the following holds with probability one:*

$$\mathbb{E}[F_t|X_{t-1}, \ldots, X_0] - F_t \leq C_{\mathrm{dev}} \cdot \alpha \cdot d^2 \cdot g_{\max} \cdot (1-\alpha)^{-t} \left(1 - \frac{\alpha}{2}\right)^t f_{\max},$$

$$\mathbf{Var}[F_t|X_{t-1}, \ldots, X_0] \leq C_{\mathrm{var}} \cdot \alpha^2 \cdot d^2 \cdot g_{\max} \cdot (1-\alpha)^{-2t} \left(1 - \frac{\alpha}{2}\right)^{2t} f_{\max}^2.$$

*Proof.* The proof is technical but straightforward; it is deferred to Section E.5. $\qquad\square$

**Lemma 9** (Function value concentration). *Let the initial point satisfy $f(X_0) \leq \frac{1}{2}f_{\max}$. Then, there exists a sufficiently small constant $c > 0$ such that for all learning rates $\alpha < c/(g_{\max}d^2 \log d)$, we have*

$$\mathbf{Pr}\left(f_i(X_{t+1})1_{\mathfrak{E}_t} > \left(1 - \frac{\alpha}{2}\right)^t f_{\max}\right) = \mathbf{Pr}\left(\mathfrak{E}_t \cap \left\{f_i(X_{t+1}) > \left(1 - \frac{\alpha}{2}\right)^t f_{\max}\right\}\right) \leq \frac{1}{2d^{10}}.$$

*Proof.* Let $\sigma^2 = \sum_{\tau=1}^{t} \mathbf{Var}[F_\tau|X_{\tau-1}, \ldots, X_0]$ and let $R$ satisfy $\mathbb{E}[F_\tau|X_{\tau-1}, \ldots, X_0] \leq X_\tau + R$ almost surely for all $\tau \in \{1, 2, \ldots, t\}$. Recall via the standard Azuma–Bernstein concetration inequality for supermartingales that $\mathbf{Pr}(F_t \geq F_0 + s) \leq \exp\left(-\frac{s^2/2}{\sigma^2 + Rs/3}\right)$. Equivalently, there exists a large enough constant $C > 0$ in $s = C \cdot (1-\alpha)^t \left[\sqrt{\sigma^2 \log d} + R \log d\right]$ such that the following is true

$$\mathbf{Pr}\left(f(X_{t+1})1_{\mathfrak{E}_t} \geq (1-\alpha)^t f(X_0) + s\right) \leq \frac{1}{2d^{10}}.$$

Given that $f(X_0) \leq \frac{1}{2}f_{\max}$ and therefore $(1-\alpha)^t f(X_0) \leq \frac{1}{2}\left(1 - \frac{\alpha}{2}\right)^t \cdot f_{\max}$ holds by hypothesis, the desired claim is true if we can show that $s \leq \frac{1}{2}\left(1 - \frac{\alpha}{2}\right)^t \cdot f_{\max}$. Crucially, we observe that the variance term in $s$ does not blow-up with time $t$

$$(1-\alpha)^{2t} \cdot \sigma^2 \leq f_{\max}^2 \cdot C_{\mathrm{var}} \cdot d^2 \cdot g_{\max} \cdot \alpha^2 \cdot \sum_{\tau=1}^{t} (1-\alpha)^{2t-2\tau} \left(1 - \frac{\alpha}{2}\right)^{2\tau}$$

$$= \left(1 - \frac{\alpha}{2}\right)^{2t} f_{\max}^2 \cdot C_{\mathrm{var}} \cdot d^2 \cdot g_{\max} \cdot \alpha^2 \cdot \sum_{\tau=1}^{t} \left(\frac{1-\alpha}{1-\alpha/2}\right)^{2t-2\tau}$$

$$\leq \left(1 - \frac{\alpha}{2}\right)^{2t} f_{\max}^2 \cdot C_{\mathrm{var}} \cdot d^2 \cdot g_{\max} \cdot \alpha$$

due to the geometric series expansion $\sum_{\tau=0}^{t} \beta^{t-\tau} = (1 - \beta^{t+1})/(1-\beta)$. Substituting the deviations term, choosing a step-size $\alpha \leq c/(\rho d^2 \log d)$ for sufficiently small $c$ yields

$$s = \left(1 - \frac{\alpha}{2}\right)^t \cdot \left[\sqrt{C_{\mathrm{var}} \cdot d^2 \cdot g_{\max} \cdot \alpha \cdot \log d} + C_{\mathrm{dev}} \cdot d^2 \cdot g_{\max} \cdot \alpha \cdot \log d\right] \cdot f_{\max}$$

$$\leq \frac{1}{2}\left(1 - \frac{\alpha}{2}\right)^t f_{\max}.$$

$\qquad\square$

We now set up a supermartingale on each of the incoherence functions $g_i$. Again, our goal is to show that the variance and the maximum deviation of this supermartingale have the right dependence on $\alpha, d, \rho, \kappa, g_{\max}$, so that a step-size of $\alpha \leq c/(\rho d^2 \log d)$ with a sufficiently small $c > 0$ will keep the cumulative deviations over $T$ iterations within a factor of 2. Note that Jin et al. [10, Section B.2]'s proof tracks a different function $h_i(X) = e_i^T X X^T e_i$ that is substantially simpler, but pays a penalty of two to three factors of the condition number $\kappa$.

**Lemma 10** (Incoherence supermartingale). *Let $g_i(X) = e_i^T X (X^T X)^{-1} X^T e_i$. Define $g_{\max} = \frac{16}{(1-2\rho)^2} \max_i g_i(Z)$. For a fixed $i \in [n]$ with sufficiently small $c > 0$, the following with learning rate $\alpha < c/(\rho d^2 \log d)$ is a supermartingale*

$$G_{it} = (1 - \zeta \cdot \alpha)^{-t} \left( g(X_t) \cdot 1_{\mathfrak{E}_{t-1}} - \frac{\zeta}{2} g_{\max} \right) \text{ where } \zeta = \frac{1 - 2\rho}{1 - \rho} < 1,$$

*meaning that $\mathbb{E}[G_{i(t+1)}|X_t, \ldots, X_0] \leq G_{it}$ holds for all $t \in \{0, 1, 2, \ldots\}$. Moreover, there exist sufficiently large constants $C_{dev}, C_{var} > 0$ with no dependence on $g_{\max}, n, t$ such that*

$$\mathbb{E}[G_{it}|X_{t-1}, \ldots, X_0] - G_{it} \leq C_{\mathrm{dev}} \cdot \alpha \cdot d^2 \cdot \rho \cdot (1 - \zeta \cdot \alpha)^{-t} g_{\max},$$

$$\mathbf{Var}[G_{it}|X_{t-1}, \ldots, X_0] \leq C_{\mathrm{var}} \cdot \alpha^2 \cdot d^2 \cdot \rho^2 \cdot (1 - \zeta \cdot \alpha)^{-2t} g_{\max}^2.$$

*Proof.* The proof is long but straightforward; it is deferred to Section E.5. $\qquad\square$

**Lemma 11** (Incoherence concentration). *Let the initial point satisfy $\max_i g_i(X_0) \leq \frac{1}{2} g_{\max}$. Then, there exists a sufficiently small constant $c > 0$ such that for all learning rates $\alpha < c/(\rho d^2 \log d)$, we have*

$$\mathbf{Pr}(g_i(X_{t+1}) 1_{\mathfrak{E}_t} > g_{\max}) = \mathbf{Pr}(\mathfrak{E}_t \cap \{g_i(X_{t+1}) > g_{\max}\}) \leq \frac{1}{2d^{11}}. \tag{21}$$

*Proof.* Let $\sigma^2 = \sum_{\tau=1}^t \mathbf{Var}[G_{i\tau}|X_{\tau-1}, \ldots, X_0]$ and let $R$ satisfy $\mathbb{E}[G_{i\tau}|X_{\tau-1}, \ldots, X_0] \leq X_\tau + R$ almost surely for all $\tau \in \{1, 2, \ldots, t\}$. Recall via the standard Azuma–Bernstein concetration inequality for supermartingales that

$$\mathbf{Pr}(G_{it} \geq G_{i0} + s) \leq \exp\left( -\frac{s^2/2}{\sigma^2 + Rs/3} \right).$$

Equivalently, there exists a large enough constant $C > 0$ such that the following is true

$$\mathbf{Pr}\left( g_i(X_{t+1}) 1_{\mathfrak{E}_t} \geq \frac{1}{2} g_{\max} + (1 - \zeta \cdot \alpha)^t \left( g(X_0) - \frac{1}{2} g_{\max} \right) + s' \right) \leq \frac{1}{2d^{11}}$$

$$\text{where } s' = C \cdot (1 - \zeta \cdot \alpha)^t \cdot \left[ \sqrt{\sigma^2 \log d} + R \log d \right].$$

Given that $g(X_0) \leq \frac{1}{2} g_{\max}$ holds by hypothesis, the desired claim is true if we can show that $s' \leq \frac{1}{2} g_{\max}$. Crucially, we observe that the variance term in $s'$ does not blow-up with time $t$

$$(1 - \zeta \cdot \alpha)^{2t} \sigma^2 \leq g_{\max}^2 \cdot C_{\mathrm{var}} \cdot d^2 \cdot \rho^2 \cdot \alpha^2 \sum_{\tau=1}^t (1 - \zeta \cdot \alpha)^{2t-2\tau}$$

$$\leq g_{\max}^2 \cdot C_{\mathrm{var}} \cdot d^2 \cdot \rho^2 \cdot \alpha$$

due to the geometric series expansion $\sum_{\tau=0}^t \beta^{t-\tau} = (1 - \beta^{t+1})/(1 - \beta)$. Substituting the deviations term, choosing a step-size $\alpha \leq c/(\rho d^2 \log d)$ for sufficiently small $c$ yields

$$s' = O\left( \sqrt{\alpha \cdot d^2 \cdot \rho^2 \cdot g_{\max}^2 \cdot \log d} \right) + O\left( \alpha \cdot d^2 \cdot \rho \cdot g_{\max} \cdot \log d \right) = \frac{g_{\max}}{2}.$$

$\qquad\square$

In summary, Lemma 9 requires a step-size of $\alpha \leq c/(g_{\max} d^2 \log d)$ to keep deviations on $f$ small, while Lemma 11 requires a step-size of $\alpha \leq c/(\rho d^2 \log d)$ to keep deviations on $g_i$ small. Therefore, it follows that a step-size $\alpha \leq c/((g_{\max} + \rho) d^2 \log d)$ will keep both deviations small.

*Proof of Theorem 2.* For a step-size $\alpha \leq c/((g_{\max} + \rho) d^2 \log d)$ with sufficiently small $c > 0$, both concentration bounds Lemma 9 and Lemma 11 are valid. Combined, we take the trivial union bound to determine the probability of *failure* at the $(t+1)$-th step, after succeeding after $t$ steps:

$$\mathbf{Pr}(\mathfrak{E}_t \cap \overline{\mathfrak{E}}_{t+1}) = \sum_{i=1}^d \mathbf{Pr}(\mathfrak{E}_t \cap \{g_i(X_{t+1}) \geq g_{\max}\}) + \mathbf{Pr}(\mathfrak{E}_t \cap \{f(X_{t+1}) \geq (1 - \frac{\alpha}{2})^{t+1} f_{\max}\}).$$

$$\leq d \cdot \frac{1}{2d^{11}} + \frac{1}{2d^{10}} = \frac{1}{d^{10}}.$$

Here, $\overline{\mathfrak{E}}_{t+1}$ denotes the complement of $\mathfrak{E}_{t+1}$. The probability of failure at the $T$-th step is then the cummulative probability of failing at the $(t+1)$-th step, after succeeding after $t$ steps, over all $t \leq T$:

$$\mathbf{Pr}(\overline{\mathfrak{E}}_T) \leq \sum_{t=1}^{T} \mathbf{Pr}(\mathfrak{E}_{t-1} \cap \overline{\mathfrak{E}}_t) \leq \frac{T}{d^{10}}$$

and this proves that $\mathfrak{E}_T$ happens with probability $1 - T/d^{10}$ as desired. $\square$

### E.5 Proofs of supermartingale deviations and variances

We will now verify the supermartingales and their deviations and variances in detail. We first begin by proving the following bounds on the size of the stochastic gradient.

**Lemma 12.** *Let $X, Z \in \mathbb{R}^{n \times r}$ satisfy $\|XX^T - ZZ^T\|_F \leq \rho \cdot \lambda_{\min}(Z^T Z)$ with $\rho < 1/2$ and $\max_i e_i^T X(X^T X)^{-1} X e_i \leq g_{\max}$ and $\max_i e_i^T Z(Z^T Z)^{-1} Z e_i \leq g_{\max}$. Then, with respect to the randomness of the following*

$$SG(X) = 2d^2 \cdot (XX^T - ZZ^T)_{i,j} \cdot (e_i e_j^T + e_j e_i^T)X$$

*where $(i,j) \sim [d]^2$ is selected uniformly at random, we have:*

1. *$\|SG(X)\|_X^* \leq 8d^2 \cdot g_{\max}^{1/2} \cdot \|XX^T - ZZ^T\|_F$.*

2. *$\|SG(X)(X^T X)^{-1}\|_X^* \leq 16d^2 \cdot g_{\max}^{1/2} \cdot \rho$.*

3. *$\mathbb{E}(\|SG(X)\|_X^*)^p \leq 2^{2p} \cdot d^{2(p-1)} \cdot g_{\max}^{p/2} \cdot \|XX^T - ZZ^T\|_F^p$.*

4. *$\mathbb{E}(\|SG(X)(X^T X)^{-1}\|_X^*)^p \leq 2^{3p} \cdot d^{2(p-1)} \cdot g_{\max}^{p/2} \cdot \rho^p$.*

*Proof.* Let us write $E = XX^T - ZZ^T$. To prove (i) we have

$$\|SG(X)\|_X^* = 2d^2 \cdot E_{i,j} \cdot (\|(e_i e_j^T + e_j e_i^T)X\|_X^*) \leq 4d^2 \cdot E_{i,j} \cdot \max_i \|e_i^T X\|_X^*$$

and if we write $Q_X = X(X^T X)^{-1/2}$ and $Q_Z = Z(Z^T Z)^{-1/2}$ we have

$$
\begin{aligned}
E_{i,j} = e_i^T E e_j &= e_i^T Q_X Q_X^T (XX^T - ZZ^T) e_j - e_i^T (I - Q_X Q_X^T) ZZ e_j \\
&\leq \|e_k^T Q_X\| \|Q_X^T (XX^T - ZZ^T) e_j\| + \|e_i^T (I - Q_X Q_X^T) ZZ^T Q_Z\| \|Q_Z^T e_j\|, \\
&\leq g_{\max}^{1/2} \cdot \|XX^T - ZZ^T\|_F + \|XX^T - ZZ^T\|_F \cdot g_{\max}^{1/2}.
\end{aligned}
$$

we use the fact that $(I - Q_X Q_X^T)(XX^T - ZZ^T) = -(I - Q_X Q_X^T) ZZ^T$ in the first and last lines. To prove (ii) we have

$$\|SG(X)(X^T X)^{-1}\|_X^* = \frac{\|SG(X)\|_X^*}{\lambda_{\min}(X^T X)} \leq \frac{8d^2 \cdot g_{\max}^{1/2} \cdot \rho \cdot \lambda_{\min}(Z^T Z)}{(1-\rho) \cdot \lambda_{\min}(Z^T Z)} \leq 16d^2 \cdot g_{\max}^{1/2} \cdot \rho$$

where we used Weyl's inequality $\lambda_r(XX^T) \geq \lambda_r(ZZ^T) - \|XX^T - ZZ^T\|_F$. To prove (iii) we have

$$\mathbb{E}(\|SG(X)\|_X^*)^p = \frac{1}{d^2} \sum_{i,j} d^{2p} \cdot (2E_{i,j})^p \cdot (\|(e_i e_j^T + e_j e_i^T)X\|_X^*)^p \leq 2^{2p} \cdot d^{2(p-1)} \cdot \|E\|_F^p \cdot \max_i (\|e_i^T X\|_X^*)^p$$

where we used $(\sum_i x_i^2)^{1/2} \geq (\sum_i x_i^p)^{1/p}$ for any $p \geq 2$. The proof of (iv) follows identically by applying the proof of (ii) to the proof of (iii). $\square$

We now prove the properties of the function value supermartingale $F_t$.

*Proof of Lemma 8.* Conditioning on the current iterate $X_t$ and the event $\mathfrak{E}_t$, the new iterate $X_{t+1} = X_t - \alpha SG(X_t)(X_t^T X_t)^{-1}$ has expectation

$$\mathbb{E}[f(X_{t+1})] \leq f(X_t) - \alpha \langle \nabla f(X_t), \mathbb{E}[SG(X_t)(X_t^T X_t)^{-1}] \rangle + \frac{L_X}{2} \alpha^2 \cdot \mathbb{E}[(\|SG(X_t)\|_X^*)^2]$$

with $L_X = O(1)$ by evoking Lemma 3 noting that $\|\alpha SG(X_t)\|_X^* = O(1) \cdot \sqrt{f(X_t)}$ for the step-size $\alpha \leq c/(g_{\max}d^2 \log d)$, since

$$\|\alpha SG(X_t)\|_X^* = \alpha \cdot 2d^2 \cdot (X_t X_t^T - ZZ^T)_{i,j} \|(e_i e_j^T + e_j e_i^T)X_t\|_X^*$$
$$\leq \alpha \cdot 2d^2 \cdot |X_t X_t^T - ZZ^T|_\infty \cdot 2\sqrt{g_{\max}}$$
$$\leq \frac{c}{g_{\max}d^2 \log d} \cdot 2d^2 \cdot \sqrt{f(X_t)g_{\max}} \cdot 2\sqrt{g_{\max}} = \frac{4c}{\log d}\sqrt{f(X_t)}.$$

The linear term evaluates simply as $\mathbb{E}[SG(X)(X^T X)^{-1}] = \nabla f(X)(X^T X)^{-1}$, while the quadratic term evaluates

$$\mathbb{E}[(\|SG(X)\|_X^*)^2] = \frac{1}{d^2}\sum_{i,j} 4d^4 \cdot (XX^T - ZZ^T)_{i,j}^2(\|(e_i e_j^T + e_j e_i^T)X\|_X^*)^2$$
$$\leq \sum_{i,j} 4d^2 \cdot (XX^T - ZZ^T)_{i,j}^2 \cdot 4g_{\max} = 16 \cdot g_{\max} \cdot d^2 \cdot f(X)$$

Combined, substituting $(\|\nabla f(X)\|_X^*)^2 \geq 13 \cdot f(X)$, it follows that we have geometric convergence

$$\mathbb{E}[f(X_{t+1})] \leq f(X_t) - \alpha\langle\nabla f(X_t), \mathbb{E}[SG(X_t)(X_t^T X_t)^{-1}]\rangle + \frac{L_X}{2}\alpha^2 \cdot \mathbb{E}[(\|SG(X_t)\|_X^*)^2]$$
$$\leq (1 - 2\alpha)f(X_t) + L_X \cdot \alpha^2 \cdot 8 \cdot g_{\max} \cdot d^2 \cdot f(X_t) \leq (1-\alpha)f(X_t)$$

where we observe that we can pick a small enough constant $c$ in the step-size $\alpha \leq c/(g_{\max}d^2 \log d)$ so that

$$L_X \cdot \alpha^2 \cdot 8 \cdot g_{\max} \cdot d^2 \cdot f(X_t) = \frac{c \cdot L_X \cdot 8 \cdot g_{\max} \cdot d^2}{g_{\max}d^2 \log d}\alpha f(X_t) \leq \alpha f(X_t).$$

Now, to confirm that $F_t$ is a martingale, it remains to see that

$$\mathbb{E}[F_{t+1}|X_t] = (1-\alpha)^{-(t+1)}\mathbb{E}[f(X_{t+1}) \cdot 1_{\mathfrak{E}_t}|X_t] \leq (1-\alpha)^{-(t+1)}(1-\alpha)f(X_t)1_{\mathfrak{E}_t}$$
$$\leq (1-\alpha)^{-t}f(X_t)1_{\mathfrak{E}_{t-1}} = F_t,$$

where the last inequality follows from $1_{\mathfrak{E}_t} \leq 1_{\mathfrak{E}_{t-1}}$.

We now bound the deviations on $F_t$. Conditioning on the previous iterates $X_t, \ldots, X_0$, we observe that the $f(X_t)$ terms cancel:

$$f(X_{t+1}) \cdot 1_{\mathfrak{E}_t} - \mathbb{E}[f(X_{t+1}) \cdot 1_{\mathfrak{E}_t}] \leq \Big[-\alpha\langle\nabla f(X_t), [SG(X_t) - \mathbb{E}SG(X_t)](X_t^T X_t)^{-1}\rangle$$
$$+ \frac{\alpha^2 \cdot L_X}{2}(\|SG(X_t)\|_X^*)^2 + \mathbb{E}(\|SG(X_t)\|_X^*)^2\Big] \cdot 1_{\mathfrak{E}_t}. \tag{22}$$

Here we have for the linear term

$$\langle\nabla f(X_t), SG(X_t)(X_t^T X_t)^{-1}\rangle \cdot 1_{\mathfrak{E}_t} \leq \|\nabla f(X_t)\|_X^*\|SG(X_t)\|_X^* \cdot 1_{\mathfrak{E}_t}$$
$$\leq 4\sqrt{f(X_t)} \cdot 4d^2\sqrt{f(X_t)} \cdot g_{\max} \cdot 1_{\mathfrak{E}_t} = O(d^2 g_{\max})f(X_t) \cdot 1_{\mathfrak{E}_t}$$

and the quadratic term

$$(\|SG(X_t)\|_X^*)^2 \cdot 1_{\mathfrak{E}_t} \leq d^4 \cdot f(X_t) \cdot g_{\max}^2 \cdot 1_{\mathfrak{E}_t} = O(d^4 g_{\max}^2)f(X_t) \cdot 1_{\mathfrak{E}_t}.$$

Therefore, using the maximum value to bound the expectation, we have

$$F_{t+1} - \mathbb{E}[F_{t+1}|X_t, \ldots, X_0] \leq \alpha(1-\alpha)^{-t} \cdot \big[O(d^2 g_{\max})f(X_t) + \alpha O(d^4 g_{\max}^2)f(X_t)\big] \cdot 1_{\mathfrak{E}_t}$$
$$\leq \alpha(1-\alpha)^{-t} \cdot O(d^2 g_{\max})f(X_t) \cdot 1_{\mathfrak{E}_t}$$
$$\leq C_{\text{dev}} \cdot \alpha(1-\alpha)^{-t}\left(1 - \frac{\alpha}{2}\right)^t f_{\max} \cdot d^2 \cdot g_{\max} \cdot 1_{\mathfrak{E}_t}$$

where again we observe that a step-size like $\alpha \leq c/(g_{\max}d^2 \log d) = O(d^{-2}g_{\max}^{-1})$ yields the cancellation of exponents $\alpha \cdot O(d^4 g_{\max}^2) = O(d^2 g_{\max})$.

Finally, we bound the variance. Conditioned on all previous iterates $X_t, \ldots, X_0$ we have

$$\mathbf{Var}(\langle \nabla f(X_t), SG(X_t)(X_t^T X_t)^{-1} \rangle \cdot 1_{\mathfrak{E}_t}) \le \mathbb{E}[\langle \nabla f(X_t), SG(X_t)(X_t^T X_t)^{-1} \rangle^2 \cdot 1_{\mathfrak{E}_t}]$$
$$\le (\|\nabla f(X_t)\|_{X_t}^*)^2 \cdot \mathbb{E}[(\|SG(X_t)\|_X^*)^2] \cdot 1_{\mathfrak{E}_t} \le O(d^2 g_{\max}) \cdot f(X_t)^2 \cdot 1_{\mathfrak{E}_t},$$

and also

$$\mathbf{Var}((\|SG(X_t)\|_X^*)^2 \cdot 1_{\mathfrak{E}_t}) \le \mathbb{E}[(\|SG(X_t)\|_X^*)^4 \cdot 1_{\mathfrak{E}_t}] = O(d^6 g_{\max}^3) f(X_t)^2 \cdot 1_{\mathfrak{E}_t}.$$

By the same expansion in (22) we have

$$\mathbf{Var}(F_{t+1}|X_t, \ldots, X_0) \le \alpha^2 (1-\alpha)^{-2t} \cdot \left[ O(d^2 g_{\max}) \cdot f(X_t)^2 + \alpha^2 O(d^6 g_{\max}^3) f(X_t)^2 \right] \cdot 1_{\mathfrak{E}_t}$$
$$\le \alpha^2 (1-\alpha)^{-2t} \cdot O(d^2 g_{\max}) \cdot f(X_t)^2 \cdot 1_{\mathfrak{E}_t}$$
$$\le C_{\mathrm{var}} \cdot \alpha^2 (1-\alpha)^{-2t} \left(1 - \frac{\alpha}{2}\right)^{2t} f_{\max}^2 \cdot d^2 \cdot g_{\max} \cdot 1_{\mathfrak{E}_t}$$

where again we observe that a step-size like $\alpha \le c/(g_{\max} d^2 \log d) = O(d^{-2} g_{\max}^{-1})$ yields the cancellation of exponents $\alpha^2 \cdot (d^6 g_{\max}^3) = O(d^2 g_{\max})$. $\square$

We now prove properties of the incoherence martingale.

*Proof of Lemma 10.* Conditioning on $X_t$ and the event $\mathfrak{E}_t$, we have for $V = SG(X_t)(X_t^T X_t)^{-1}$

$$\mathbb{E}[g_i(X_{t+1})] \le g_k(X_t) - \alpha \langle \nabla g_i(X_t), \mathbb{E}[V] \rangle + \alpha^2 \cdot \mathbb{E}\left[\frac{(\|V\|_X^*)^2}{1 - 2\|V\|_X^*}\right]$$
$$\le \left(1 - \frac{1 - 2\rho}{1 - \rho}\alpha\right) g_i(X_t) + \alpha \cdot \frac{1}{1-\rho} \cdot \sqrt{g_i(X) g_i(Z)} + \alpha^2 \cdot \frac{\mathbb{E}\left[(\|V\|_X^*)^2\right]}{1 - 2\|V\|_X^*}$$
$$\le \left(1 - \frac{1 - 2\rho}{1 - \rho}\alpha\right) g_i(X_t) + \alpha \cdot \frac{\sqrt{g_i(Z)/g_{\max}}}{1-\rho} \cdot g_{\max} + \alpha^2 \cdot \frac{O(d^2 \cdot g_{\max} \cdot \rho^2)}{1 - O(g_{\max}^{1/2} \cdot \rho)}$$
$$\le (1 - \zeta\alpha) g_i(X_t) + \alpha \cdot \frac{\zeta}{2} g_{\max} \qquad \text{for } \alpha = O(\rho^{-1} d^{-2}).$$

Here we note that we have carefully chosen $g_{\max}$ so that the ratio $\max_i g_i(Z)/g_{\max} = [(1-\rho) \cdot \zeta/4]^2$. It then follows that the following is a supermartingale

$$G_{it} = (1 - \zeta\alpha)^{-t} \left(g_i(X_t) \cdot 1_{\mathfrak{E}_{t-1}} - \frac{\zeta}{2} g_{\max}\right).$$

Indeed, we have

$$\mathbb{E}[G_{i(t+1)}|X_t, \ldots, X_0] = (1 - \zeta\alpha)^{-(t+1)} \left(\mathbb{E}[g_i(X_{t+1}) \cdot 1_{\mathfrak{E}_t}|X_t, \ldots, X_0] - \frac{\zeta}{2} g_{\max}\right)$$
$$\le (1 - \zeta\alpha)^{-(t+1)} \left[(1 - \zeta\alpha) g_i(X_t) \cdot 1_{\mathfrak{E}_t} + \alpha \cdot \frac{\zeta}{2} g_{\max} \cdot 1_{\mathfrak{E}_t} - \frac{\zeta}{2} g_{\max}\right]$$
$$\le (1 - \zeta\alpha)^{-t} \left[g_i(X_t) \cdot 1_{\mathfrak{E}_{t-1}} - \left(\frac{1-\alpha}{1-\alpha}\right) \cdot \frac{\zeta}{2} g_{\max}\right] = G_{it}$$

where the final line uses $1_{\mathfrak{E}_t} \le 1_{\mathfrak{E}_{t-1}} \le 1$.

We now bound the deviations on $G_{it}$. Conditioning on the previous iterates $X_t, \ldots, X_0$, we obseve that the $g_i(X_t)$ terms cancel:

$$g_i(X_{t+1}) \cdot 1_{\mathfrak{E}_t} - \mathbb{E}[g_i(X_{t+1}) \cdot 1_{\mathfrak{E}_t}] \le \left[-\alpha \langle \nabla g_i(X_t), [SG(X_t) - \mathbb{E}SG(X_t)](X_t^T X_t)^{-1} \rangle \right.$$
$$\left. + 5\alpha^2 \cdot \frac{(\|SG(X_t)(X_t^T X_t)^{-1}\|_X^*)^2 + \mathbb{E}(\|SG(X_t)(X_t^T X_t)^{-1}\|_X^*)^2}{1 - \alpha \|SG(X_t)(X_t^T X_t)^{-1}\|_X^*}\right] \cdot 1_{\mathfrak{E}_t}. \quad (23)$$

Here we have for the linear term

$$\|\nabla g(X)\| \cdot 1_{\mathfrak{E}_t} \le \|2[I - X(X^T X)^{-1} X^T] e_i e_i^T X(X^T X)^{-1}\| \cdot 1_{\mathfrak{E}_t} \le O(\kappa^{1/2} \cdot g_{\max}^{1/2})$$

and $\|SG(X)\|_X^* \cdot 1_{\mathfrak{E}_t} \le O(d^2 \cdot \sqrt{g_{\max} f(X)}) = O(d^2 \sqrt{g_{\max}} \rho/\kappa)$ noting that $f_{\max} = \rho^2/\kappa^2$ and hence

$$\begin{aligned} |\langle \nabla g(X_t), SG(X_t)(X_t^T X_t)^{-1} \rangle| &\le \|\nabla g(X_t)\| \cdot \|SG(X_t)\|_X^* \\ &\le O(\sqrt{\kappa g_{\max}}) \cdot O(d^2 \sqrt{g_{\max}} \rho/\kappa) = O(g_{\max} \cdot d^2 \rho) \cdot 1_{\mathfrak{E}_t}. \end{aligned}$$

We have for the quadratic term

$$\frac{5(\|SG(X_t)(X_t^T X_t)^{-1}\|_X^*)^2}{1 - \alpha \|SG(X_t)(X^T X)^{-1}\|_X^*} \cdot 1_{\mathfrak{E}_t} \le \frac{O(d^4 \cdot g_{\max} \cdot \rho^2)}{1 - \alpha \cdot O(d^2 \cdot g_{\max}^{1/2} \cdot \rho)} \cdot 1_{\mathfrak{E}_t} = O(g_{\max} \cdot d^4 \rho^2) \cdot 1_{\mathfrak{E}_t}.$$

Therefore, using the maximum value to bound the expectation, we have

$$\begin{aligned} G_{i(t+1)} - \mathbb{E}[G_{i(t+1)}|X_t, \dots, X_0] &\le \alpha (1-\alpha)^{-t} g_{\max} \cdot \left[ O(d^2 \rho) + \alpha O(d^4 \rho^2) f(X_t) \right] \cdot 1_{\mathfrak{E}_t} \\ &\le \alpha (1-\alpha)^{-t} g_{\max} \cdot O(d^2 \rho) \cdot 1_{\mathfrak{E}_t} \end{aligned}$$

where again we observe that a step-size like $\alpha \le c/(\rho d^2 \log d) = O(d^{-2} \rho^{-1})$ yields the cancellation of exponents $\alpha \cdot O(d^4 \rho^2) = O(d^2 \rho)$.

Finally, we bound the variance. Conditioned on all previous iterates $X_t, \dots, X_0$ we have

$$\begin{aligned} \mathbf{Var}(\langle \nabla g_i(X_t), SG(X_t)(X_t^T X_t)^{-1} \rangle \cdot 1_{\mathfrak{E}_t}) &\le \mathbb{E}[\langle \nabla g_i(X_t), SG(X_t)(X_t^T X_t)^{-1} \rangle^2 \cdot 1_{\mathfrak{E}_t}] \\ &\le (\|\nabla g_i(X_t)\|)^2 \cdot \mathbb{E}[(\|SG(X_t)\|_X^*)^2] \cdot 1_{\mathfrak{E}_t} \le O(g_{\max}^2 \cdot d^2 \rho^2) \cdot 1_{\mathfrak{E}_t}, \end{aligned}$$

and also

$$\begin{aligned} \mathbf{Var}\left( \frac{5(\|SG(X_t)(X_t^T X_t)^{-1}\|_X^*)^2}{1 - \alpha \|SG(X_t)(X_t^T X_t)^{-1}\|_X^*} \cdot 1_{\mathfrak{E}_t} \right) &\le \frac{25 \cdot \mathbb{E}[(\|SG(X_t)(X_t^T X_t)^{-1}\|_X^*)^4]}{(1 - \alpha \|SG(X_t)(X_t^T X_t)^{-1}\|_X^*)^2} \cdot 1_{\mathfrak{E}_t} \\ &\le \frac{O(g_{\max}^2 \cdot d^6 \rho^4)}{1 - \alpha \cdot O(d^2 \cdot g_{\max}^{1/2} \cdot \rho)} \cdot 1_{\mathfrak{E}_t} = O(g_{\max}^2 \cdot d^6 \rho^4) \cdot 1_{\mathfrak{E}_t}. \end{aligned}$$

By the same expansion in (23) we have

$$\begin{aligned} \mathbf{Var}(G_{i(t+1)}|X_t, \dots, X_0) &\le \alpha^2 (1-\alpha)^{-2t} \cdot \left[ O(g_{\max}^2 \cdot d^2 \rho^2) \cdot 1_{\mathfrak{E}_t} + \alpha^2 O(g_{\max}^2 \cdot d^6 \rho^4) \cdot 1_{\mathfrak{E}_t,} \right] \cdot 1_{\mathfrak{E}_t} \\ &\le \alpha^2 (1-\alpha)^{-2t} g_{\max}^2 \cdot O(d^2 \rho^2) \cdot 1_{\mathfrak{E}_t} \end{aligned}$$

where again we observe that a step-size like $\alpha \le c/(\rho d^2 \log d) = O(d^{-2} \rho^{-1})$ yields the cancellation of exponents $\alpha^2 \cdot (d^6 \rho^4) = O(d^2 \rho^2)$. $\qquad \square$