# OpenReview forum: "Accelerating SGD for Highly Ill-Conditioned Huge-Scale Online Matrix Completion"
_NeurIPS.cc/2022/Conference — NeurIPS 2022 Accept_

### Official Review · Reviewer_To6Q · 2022-06-27

**Rating:** 5
**Confidence:** 4
**Soundness:** 2 fair
**Presentation:** 3 good
**Contribution:** 2 fair

**Summary:**

This paper propose new online algorithm for matrix completion. When the underlying low-rank matrix is ill-conditioned (has a large condition numer), the authors delivered improved theoretical guarantees for the local convergence of the algorithm.

**Questions:**

The claim on line 156-158 is not correct. Full-batch gradient descent also does not require regularizer or projection to maintain incoherence, which was shown in [5].

[5] Ma, C., Wang, K., Chi, Y., & Chen, Y. (2020). Implicit Regularization in Nonconvex Statistical Estimation: Gradient Descent Converges Linearly for Phase Retrieval, Matrix Completion, and Blind Deconvolution. Foundations of Computational Mathematics, 20(3), 451-632.

**Limitations:**

This is a theory paper and I believe there is no potential negative societal impact.

**Strengths And Weaknesses:**

Strengths: This paper makes solid contribution to online matrix completion when the ground truth is ill-conditioned. The authors present a new algorithm that provably handles ill-conditioned low-rank matrix with theoretical guarantees. Extensive numerical experiments are conducted to illustrate the theory.

Weakness: my concerns for this paper is mainly about its novelty, as discussed in the first point below.
1. Given that prior literature [1] already showed the fast convergence of SGD for online matrix completion with well-conditioned ground truth, and a series of recent work [2,3,4] showing that scaling provably improves the performance of gradient descent in the presence of ill-conditioned ground truth, it is not surprising to see that by incorporating the scaling scheme with SGD will improve the performance for online matrix completion with ill-conditioned ground truth. What is the technical novelty of this paper? I recommend the authors to highlight their technical contributions of this paper.
2. The theory in this paper only guarantees local convergence. If I am not missing anything, this paper does not discuss how to obtain an initialization that satisfies the conditions in Theorem 2. The most popular initialization scheme is spectral method. Under what conditions does spectral method or other method satisfy the conditions in Theorem 2? How does these conditions depend on the condition number? If we don't have a good initialization scheme when the ground truth is ill-conditioned, then the local convergence result in this paper alone will not be so useful.
3. The claim that
4. The writing quality of this paper need to be improved, and the paper need to be proof-read. For example, only on page 4 I see the following problems:
(a) the equation (SGD) contains multiple typos;
(b) the sentence "many iterations of SGD should concentrate about the behavior of full-batch gradient descent" is ungrammatical;
(c) on line 140, "a cubic factor of $\kappa^3$" is actually $\kappa^9$;
(d) on line 144 "initial" -> "initialization" or "initial point";
(e)  on line 145 "with at least XXX probability" -> "with probability at least XXX"

[1] Jin, C., Kakade, S. M., & Netrapalli, P. (2016). Provable efficient online matrix completion via non-convex stochastic gradient descent. Advances in Neural Information Processing Systems, 29.

[2] Tong, T., Ma, C., & Chi, Y. (2021). Accelerating Ill-Conditioned Low-Rank Matrix Estimation via Scaled Gradient Descent. J. Mach. Learn. Res., 22, 150-1.

[3] Tong, T., Ma, C., & Chi, Y. (2021). Low-rank matrix recovery with scaled subgradient methods: Fast and robust convergence without the condition number. IEEE Transactions on Signal Processing, 69, 2396-2409.

[4] Tong, T., Ma, C., Prater-Bennette, A., Tripp, E., & Chi, Y. (2022, May). Scaling and Scalability: Provable Nonconvex Low-Rank Tensor Completion. In International Conference on Artificial Intelligence and Statistics (pp. 2607-2617). PMLR.

---------------After rebuttal----------------------

I would like to thank the authors for addressing my comments. I have updated the score accordingly.

---

> ### Author Response · Authors · 2022-08-01
> **Response to Reviewer To6Q**
>
> Thank you for your thoughtful comments and suggestions. Please see
> our response to your concerns below.
> > Given that prior literature [1] already showed the fast convergence
> of SGD for online matrix completion with well-conditioned ground truth,
> and a series of recent work [2,3,4] showing that scaling provably
> improves the performance of gradient descent in the presence of ill-conditioned
> ground truth, it is not surprising to see that by incorporating the
> scaling scheme with SGD will improve the performance for online matrix
> completion with ill-conditioned ground truth. What is the technical
> novelty of this paper? I recommend the authors to highlight their
> technical contributions of this paper.
>
> We thank the reviewer for this suggestion. We fully agree with the
> reviewer that it is natural and unsurprising that applying the preconditioner
> $(X^{T}X)^{-1}$ should result in faster convergence in expectation.
> Instead, we clarify that the surprising finding of our work is actually
> that the same preconditioner is also able to provide substantial *variance
> reduction* to the iterates. In fact, it is this effect of variance
> reduction that is responsible for most ($\kappa^{3}$ out of $\kappa^{4}$)
> of our improvement over the existing state-of-the-art. Our main theoretical
> contribution in this paper is a rigorous analysis of this mechanism.
>
> To explain, the core challenge of stochastic algorithms like SGD over
> deterministic algorithms like GD is that they introduce substantial
> *variance* that can "drown out" the progress made in each
> iteration. Now, a rough analysis would suggest that incorporating
> the highly ill-conditioned preconditioner should improve the convergence
> rate of the sequence in expectation, but at the cost of dramatically
> worsening the variance. The net effect should be an uncompetitive
> algorithm in the stochastic setting. For example, reducing the step-size
> to control the variance would "improve" the $\tilde{O}(\kappa^{4}d)$
> iteration count of SGD to $O(d^{2})$, which is technically independent
> of $\kappa$, but vacuous as it requires examining all $d^{2}$ elements
> of the underlying matrix.
>
> Surprisingly, we find in this paper that the specific scaling $(X^{T}X)^{-1}$
> used in ScaledSGD not only does not worsen the variance, but in fact
> *improves* it. As summarized by Reviewer XHQZ, our key insight
> and main contribution is Lemma 4, which shows that ScaledSGD obeys
> a "descent lemma" for the coherence of the iterate, with a linear
> convergence rate that is independent of $\kappa$. Whereas previously
> Jin et al. showed that SGD needs to use a step-size of $O(\kappa^{-1})$
> to keep the coherence and therefore variance at $O(\kappa^{2})$,
> our analysis based on Lemma 4 shows that ScaledSGD is able to use
> a step-size of $O(1)$ to keep the coherence and variance at $O(1)$.
> The same preconditioning mechanism that allows ScaledGD to converge
> faster than regular GD (as in Lemma 3) turns out to also allow its
> stochastic variant ScaledSGD to enjoy reduced variance compared to
> regular SGD (as in Lemma 4).
>
> Indeed, the above analysis shows variance reduction provides a factor
> of $\kappa^{3}$ improvement, and that it is actually only the final
> factor of $\kappa$ that is due to the faster convergence in expectation.
> To the best of our knowledge, this is the *first* work that demonstrates
> variance reduction through the use of a carefully-chosen preconditioner,
> as opposed to the use of "aggregate" random variables and carefully-chosen
> step-sizes, as is the norm in previous work (see e.g. [R1,R2]).
>
>
> **References**
>
> [R1] Cutkosky, Ashok, and Francesco Orabona. "Momentum-based variance reduction in non-convex sgd." Advances in neural information processing systems 32 (2019).
>
> [R2] Gorbunov, Eduard, Filip Hanzely, and Peter Richtárik. "A unified theory of SGD: Variance reduction, sampling, quantization and coordinate descent." International Conference on Artificial Intelligence and Statistics. PMLR, 2020.

---

> ### Author Response · Authors · 2022-08-01
> **Response to Reviewer To6Q (continued)**
>
> > The theory in this paper only guarantees local convergence. If I am
> not missing anything, this paper does not discuss how to obtain an
> initialization that satisfies the conditions in Theorem 2. The most
> popular initialization scheme is spectral method. Under what conditions
> does spectral method or other method satisfy the conditions in Theorem
> 2? How does these conditions depend on the condition number? If we
> don't have a good initialization scheme when the ground truth is ill-conditioned,
> then the local convergence result in this paper alone will not be
> so useful.
>
> We thank the reviewer for bringing up this point. We would like to
> respond by drawing a parallel to Newton's method, which in theory
> has an extremely narrow local region where it converges at a superlinear
> rate. One could also criticize Newton's method in that "if we don't
> have a good initialization scheme then the local convergence result
> will not be so useful." Nevertheless, it remains important to understand
> the *underlying mechanism* by which Newton's method is able to
> achieve superlinear convergence. In turn, the same mechanism can then
> be replicated in more competitive variants of Newton's method, including
> quasi-Newton methods like L-BFGS, and globalized variants like cubic
> regularized Newton and trust-region Newton.
>
> We view our contributions along the same lines. The purpose of our
> theoretical analysis is to shed light on the underlying mechanism
> by which ScaledSGD is able to achieve $\kappa$-independent local
> convergence. As described above, our key insight is that a careful
> choice of preconditioner can be used as a mechanism for *variance
> reduction*, while at the same time also fulfilling its usual, classical
> purpose, which is to accelerate convergence in expectation. Within
> the stochastic context, it turns out that it is the variance reduction
> that provides most of the acceleration, and not the improved convergence
> rate in expectation.
>
> Naturally, the same mechanism can be replicated in more practical
> variants of ScaledSGD. For example, our analysis can be repeated largely
> verbatim using the PrecGD preconditioner $(X^{T}X+\eta I)^{-1}$.
> Based on recent results for the full-batch, deterministic setting,
> we believe the resulting PrecSGD should enjoy a much wider region
> of local convergence. Also, we note that it may be possible to apply
> a preconditioner to the spectral initialization itself (by computing
> the eigenvectors for a matrix pencil of the form $M-\lambda P$ where
> $P$ is the preconditioner) in order to fully eliminate all dependencies
> of $\kappa$, including in the sample complexity. However, once the
> variance reduction mechanism has been established in the present paper,
> we believe that the above should quickly follow as natural extensions.
>
> Finally, we emphasize that our work also has important practical implications,
> beyond a purely theoretical result. Matrix factorization models are
> a ubiquitous component of real-world collaborative filtering systems.
> In practice, random initialization is routinely used to train such
> models without issue, but slow convergence (and resulting poor prediction
> quality) is readily and consistently observed for even mildly ill-conditioned
> matrices. Our work offers a practical remedy for ill-conditioning;
> we support our proposal with both a rigorous theoretical analysis
> as well as extensive 10/31 pages of experimental validation. Our hope
> is to make a compelling case for practitioners to re-train their own
> matrix factorization models in their own applications.
>
> > The writing quality of this paper need to be improved, and the paper
> need to be proof-read.
>
> We apologize for the typos and thank the reviewer for pointing them
> out. We endeavor to carefully and thoroughly proofread, and correct
> these typos in the revised version of our paper.
>
> > The claim on line 156-158 is not correct. Full-batch gradient descent
> also does not require regularizer or projection to maintain incoherence,
> which was shown in [5].
>
> We thank the reviewer for the reference. The reviewer is right; gradient
> descent does indeed maintain incoherence, as shown in [5]. Our
> claim on line 156-158 had only intended to contrast our proposed method
> with its full-batch version, ScaledGD, which does indeed require an
> explicit projection for matrix completion. We will update our paper
> to include the reviewer's reference and make the point clearer that
> it is only ScaledGD that requires an explicit projection step.

---

### Official Review · Reviewer_3eem · 2022-07-11

**Rating:** 7
**Confidence:** 4
**Soundness:** 3 good
**Presentation:** 3 good
**Contribution:** 3 good

**Summary:**

The matrix completion is a classic type of method that tries to recover the ground truth from the observation elements. The stochastic gradient descent (SGD) is one of the solutions to the huge-scale matrix completion in the real world. However, it is suffering from the ill-conditioned ground truth. This work is proposing an SGD-based method that is not affected by the ill-conditioned ground truth. Meanwhile, achieves the per iteration complexity improvement.

**Questions:**


For soundness, it could be better to emphasize whether or not the algorithm is trying to invert the ill-conditioned matrix and the XX^T for complexity discussion.

**Limitations:**

No potential negative societal impact.

**Strengths And Weaknesses:**

Strengths：
The SGD version in this problem is missing, and this work is trying to fill the missing piece.

This work is a nice extension of previous work, including the full-gradient method ScaledGD and PrecGD, the matrix completion SGD results (symmetric PSD case) from Jin [40]. The main contribution of this work could be eliminating the k-dependency in the complexity.

The experiment in the paper is synthetic data, but they offer real data in supplementary.

It’s a solid improvement on per iteration complexity.

Weaknesses：
It’s not actually a weakness. The main contribution is about eliminating the condition number k in complexity. It reads like the previous work did not consider that at all. However, the previous work ScaledGD did discuss the case that k is independent. It’s fine but probably fairer to mention that to make it clearer.

---

> ### Author Response · Authors · 2022-08-01
> **Response to Reviewer 3eem**
>
> Thank you for your thoughtful comments. Please see our response to
> your comments below.
> > The main contribution is about eliminating the condition number $\kappa$
> in complexity. It reads like the previous work did not consider that
> at all. However, the previous work ScaledGD did discuss the case that
> k is independent. It's fine but probably fairer to mention that to
> make it clearer.
>
> Thank you for pointing this out. We mention in the introduction (lines
> 55-56): "This paper is inspired by a recent full-batch gradient
> method called ScaledGD [3, 8] and PrecGD [4] in
> which gradient descent is made immune to ill-conditioning in the ground
> truth". However, we will revise our introduction to make
> the point clearer that previous work had already considered eliminating
> the condition number.
> > For soundness, it could be better to emphasize whether or not the
> algorithm is trying to invert the ill-conditioned matrix and the $XX^{T}$
> for complexity discussion.
>
> In each iteration of our algorithm, we need to multiply with $(X_{t}^{T}X_{t})^{-1}$,
> where $X_{t}$ is the $d\times r$ current iterate. Directly forming
> $X_{t}^{T}X_{t}$ would cost $O(dr^{2})$ time, and directly inverting
> this ill-conditioned matrix would cause issues with numerical stability.
> Instead, we note that SGD updates only two rows of $X_{t}$ each iteration,
> so we can implicitly carry (or cache) the preconditioner matrix $P=(X_{t}^{T}X_{t})^{-1}$,
> and then update it using the Sherman-Morrison rank-1 update formula
> four times after each $t$-th iteration (see equation (2b)). This
> allows us to avoid explicitly forming $(X_{t}^{T}X_{t})^{-1}$ while
> reducing the computational complexity of computing the preconditioner
> to $O(r^{2})$. The surprising numerical stability of this procedure
> is closely reminescent of other methods that use the Sherman-Morrison
> to update a matrix inverse, most notably in the BFGS update formula.
>
> As you suggested, we will revise our paper to incorporate the above
> into our discussions.
>
>
>
> **References**
>
> [3] Tian Tong, Cong Ma, Ashley Prater-Bennette, Erin Tripp, and Yuejie Chi. Scaling and scalability: Provable nonconvex low-rank tensor completion. In International Conference on Artificial Intelligence and Statistics, pages 2607–2617. PMLR, 2022.
>
> [4] Jialun Zhang, Salar Fattahi, and Richard Y Zhang. Preconditioned gradient descent for overparameterized nonconvex matrix factorization. Advances in Neural Information Processing Systems, 34:5985–5996, 2021.
>
> [8] Tian Tong, Cong Ma, and Yuejie Chi. Accelerating ill-conditioned low-rank matrix estimation via scaled gradient descent. Journal of Machine Learning Research, 22(150):1–63, 2021.

---

> > ### Comment · Reviewer_3eem · 2022-08-07
> > **Response to the author**
> >
> > Thank you for the response.
> > My concerns were resolved and I have no further concerns.

---

### Official Review · Reviewer_XHQZ · 2022-07-13

**Rating:** 6
**Confidence:** 2
**Soundness:** 3 good
**Presentation:** 3 good
**Contribution:** 2 fair

**Summary:**

The paper looks at online matrix-completion -- the goal is to obtain a matrix $X \in \mathbb{R}^{d \times r}$ that minimizes $\|XX^\top - M\| _F^2$ when the elements of the $d \times d$ matrix $M$ are revealed in an online fashion. The paper looks at improving the convergence guarantees of SGD for this task, in terms of the dependence in the condition number $\kappa$ of the ground-truth $M$.

A previous result of [1] showed a convergence guarantee of $O(\kappa^4 dr\log(d/\epsilon))$ for the regular SGD update. This result shows that multiplying the stochastic gradient with a pre-conditioner -- $P = (XX^\top)^{-1}$ -- gives the aforementioned convergence guarantee without any dependence on $\kappa$.

The paper shows that this preconditioner can be maintained by making 4 calls to the Sherman-Morris formula and incurs an additional cost of $O(r^2)$ as compared to $O(r)$ in regular SGD.

The main contribution in the analysis is a descent lemma -- arguing that the coherence of the iterates remains $O(1)$ throughout.

[1] Chi Jin, Sham M Kakade, and Praneeth Netrapalli. Provable efficient online matrix completion via non-convex stochastic gradient descent. Advances in Neural Information Processing Systems, 29, 2016.


**Questions:**

Since you require $\|X_0X_0^\top - ZZ^\top\|_F^2 < 1/(\kappa \cdot d)$, which is a $1/d$ factor smaller than required for [1], couldn't you make the critique that, for a fair comparison to [1], your result in fact needs an $O(d)$ **factor** more iterations? If that's the case then the complexity of ScaledSGD is $~O(d^2)$, which as pointed out is vacuous. Am I missing something here?

**Limitations:**

Yes.

**Strengths And Weaknesses:**

SGD for Online matrix-completion seems to be well-motivated [1] and applying a pre-conditioner in the descent step is a natural idea that also seems to be well-studied for full GD [2,3]. Hence, this idea is natural but the main contribution seems to be in the analysis -- specifically in Lemma 4 -- that argues that the coherence of the iterates decreases at a linear rate, independent of $\kappa$. This analysis seems to be an important contribution to understanding the preconditioned stochastic optimization for RMSE matrix-completion.

WEAKNESS

As you have mentioned. It seems like, the quality of approximation required for the initial iterate $X_0$ contains a factor of $1/d$ that [1] doesn't. Please see questions section.


[2] Jialun Zhang, Salar Fattahi, and Richard Y Zhang. Preconditioned gradient descent for over329
parameterized nonconvex matrix factorization. Advances in Neural Information Processing
330 Systems, 34:5985–5996, 2021.

[3] Tian Tong, Cong Ma, and Yuejie Chi. Accelerating ill-conditioned low-rank matrix estimation
341 via scaled gradient descent. Journal of Machine Learning Research, 22(150):1–63, 2021.

---

> ### Author Response · Authors · 2022-08-01
> **Response to Reviewer XHQZ**
>
> Thank you for your thoughtful comments and helpful question. Please
> see our response below.
> > Since you require $\|X_{0}X_{0}^{\top}-ZZ^{\top}\|_{F}^{2}<1/(\kappa\cdot d)$,
> which is a $1/d$ factor smaller than required for [1], couldn't
> you make the critique that, for a fair comparison to [1], your
> result in fact needs an $O(d)$ factor more iterations? If that's
> the case then the complexity of ScaledSGD is $O(d^{2})$, which as
> pointed out is vacuous. Am I missing something here?
>
> Indeed, our algorithm requires a smaller initial neighborhood compared
> to [1]. For a fair comparison with [1], both methods should
> start at the *same* initial neighborhood. Specifically, for
> the same radius $R/d$, [1] says that SGD converges to an $\epsilon$-accurate
> solution in $O(\kappa^{4}d\log(d)\log(R/(d\epsilon))$ iterations,
> while our paper says that ScaledSGD converges in $O(d\log(d)\log(R/(d\epsilon))$
> iterations. For any $R$ for which Theorem 2 is valid, ScaledSGD is
> still rigorously faster by a factor of $\kappa^{4}$.

---

> > ### Comment · Reviewer_XHQZ · 2022-08-08
> > **Clarifying iteration complexity**
> >
> > Thank you for this clarifying this.
> >
> > I understand that that there's better convergence for ScaledSGD over SGD when $X_0X_0^\top$ is within an $1/(\kappa \cdot d)$ radius. Is it easy to how to obtain such an $X_0$ using $O(d\log^2(d))$ iterations? Say, with constant probability success.

---

> > > ### Author Response · Authors · 2022-08-09
> > > **Response to clarifying iteration complexity**
> > >
> > > The reviewer asks whether it is still "easy" to compute an initial
> > > point within an $1/(\kappa\cdot d)$ radius, in view of the extra
> > > factor of $1/d$. If we define "easy" in terms of theoretical
> > > complexity in dimension $d$ while ignoring the condition number $\kappa$,
> > > then the answer is "yes, it is easy". Assume coherence $\mu=O(1)$
> > > and rank $r=O(1)$ for simplicity, and consider the following scheme:
> > > * (Coarse initialization) We use the spectral method to find an initial
> > > point within an $1/\kappa$ radius.
> > > * (Refinement) We use regular SGD to refine the initial point until
> > > it is within an $1/(\kappa\cdot d)$ radius.
> > >
> > > According to Jin et al., the first step uses $O(\kappa^{2}\cdot d\log d)$
> > > samples and time, and the second step uses $O(\kappa^{4}\cdot d\log^{2}d)$
> > > time. If we substitute $\kappa=O(1)$, then we indeed obtain an initial
> > > point within a $1/(\kappa\cdot d)$ radius using $O(d\log d)$ samples
> > > and in $O(d\log^{2}d)$ time.
> > >
> > > The refinement step in the above can be improved. If we use full-batch
> > > ScaledGD instead of SGD, then according to Tong et al., the second
> > > step uses $O(d\log^{2}d)$ time, with no dependence on $\kappa$.
> > > The overall initialization scheme now uses $O(\kappa^{2}\cdot d\log^{2}d)$
> > > samples and time.
> > >
> > > The above sketches show that our work gives a (small) strict theoretical improvement
> > > over the previous state-of-the-art, namely SGD via Jin et al. and
> > > the full-batched ScaledGD via Tong et al. But we emphasize
> > > that in practice, the spectral method is not actually "easy" to
> > > implement in the huge-scale setting, because it requires taking numerous inner products over length-$d$ vectors, where $d$ could be something like 100 million. On the other hand, a random initialization is "practically easy"
> > > and works very well, because the local region of convergence tends
> > > to be much larger than predicted by theory. The situation is very
> > > comparable to Newton's method, which also enjoys a much larger local
> > > region than rigorously justifiable by theory.
> > >
> > > Finally, one may ask: it is possible to initialize with no dependence
> > > on $\kappa$? We believe the answer is "fundamentally no", irrespective
> > > of the extra factor of $1/d$. On one hand, an initial radius of $1/\kappa$
> > > is necessary to exclude saddle points (obtained by trucating the $r$-th
> > > smallest singular value). On the other hand, if it is possible to
> > > initialize within a radius of $1/\kappa$ in $O(1)$ time, independent
> > > of $\kappa$, then it should be possible to engineer $\kappa$ (by
> > > overparameterizing $r$ and adding noise) to compute an $\epsilon$-accurate
> > > solution, with no time dependence on $\epsilon$. This latter implication
> > > does not make sense to us.

---

### Author Response · Authors · 2022-08-01
**Overall Clarification on Contributions**

We thank the reviewers for their helpful comments, questions, and
suggestions. To begin, we make some general comments to clarify the
practical aspect of our contributions. We then respond to each reviewer
below.

Matrix factorization models are a ubiquitous component in collaborative
filtering systems, which are used to determine what news and ads we
see, what media we consume, what we buy, and even who we date. For
such models, slow convergence (and resulting poor prediction quality)
is readily and consistently observed for even mildly ill-conditioned
matrices. Within the huge-scale setting, the quality of a matrix factorization
model cannot be benchmarked by plotting against a "ground truth".
It is possible that existing models---potentially already deployed
out in the field---currently make much worse predictions than what
they are capable of, simply due to SGD's inability to train them in
the presence of ill-conditioning.

Our proposed ScaledSGD offers a practical remedy for ill-conditioning;
it allows the same real-world models to be *re-trained* using
essentially the same infrastructure as SGD. In order to make a convincing
case to practitioners, *we dedicate 10/31 pages of this paper
to experimental validation*. In particular, we train a collaborative
filter in accordance with industry best practices (i.e. using a pairwise
ranking loss instead of RMSE) on a real-world dataset (i.e. the complete
MovieLens dataset) with 25 million pairwise comparisons. The improvement
that we observed in ScaledSGD over SGD makes a compelling case for
practitioners to re-train their own matrix factorization models in
their own applications.

---

### Meta-Review · Area_Chair_avRZ · 2022-08-30

**Recommendation:** Accept
**Confidence:** Certain

**Metareview:**

The SGD version for this matrix completion problem is missing, and this work is trying to fill the missing piece.
The concerns of the reviewers seem to be resolved.
Please try to add the real-data experiments to the final version, maybe instead of the synthetic data.

**Award:**

No

---

### Decision · Program_Chairs · 2022-09-14

Accept